# Efficient Spectral Control of Partially Observed Linear Dynamical Systems

**Anand Brahmbhatt**[1*]   **Gon Buzaglo**[1*]   **Sofiia Druchyna**[1*]   **Elad Hazan**[1,2*]

[1]Computer Science Department, Princeton University
[2]Google DeepMind Princeton

## Abstract

We propose a new method for the problem of controlling linear dynamical systems under partial observation and adversarial disturbances. Our new algorithm, Double Spectral Control (DSC), matches the best known regret guarantees while exponentially improving runtime complexity over previous approaches in its dependence on the system's stability margin. Our key innovation is a two-level spectral approximation strategy, leveraging double convolution with a universal basis of spectral filters, enabling efficient and accurate learning of the best linear dynamical controllers.

## 1 Introduction

Control theory is a decades-old branch of applied mathematics concerned with designing systems that maintain desirable behavior over time, with applications ranging from robotics and aerospace to economics and biology. Recently, it has been adopted by the machine learning community through the lens of online learning, enabling new approaches to sequential decision making in systems with latent state and feedback.

A central model in control theory, and increasingly in reinforcement learning and sequence prediction, is the linear dynamical system (LDS), where the hidden state $\mathbf{x}_t \in \mathbb{R}^d$ evolves linearly in response to control inputs $\mathbf{u}_t \in \mathbb{R}^n$ and adversarial disturbances $\mathbf{w}_t$, and only partial observations $\mathbf{y}_t \in \mathbb{R}^p$ are available:

$$
\begin{aligned}
\mathbf{x}_{t+1} &= A\mathbf{x}_t + B\mathbf{u}_t + \mathbf{w}_t \,, \\
\mathbf{y}_t &= C\mathbf{x}_t \,.
\end{aligned}
\tag{1}
$$

where $\mathbf{u}_t \in \mathbb{R}^n$ is the control input, $\mathbf{w}_t \in \mathbb{R}^d$ is an adversarial disturbance, and $\mathbf{y}_t \in \mathbb{R}^p$ is a partial observation of the latent state.

At each round $t$, the learner observes $\mathbf{y}_t$, selects a control $\mathbf{u}_t$, and incurs a convex loss $c_t(\mathbf{y}_t, \mathbf{u}_t)$, where the loss functions $c_t$ are chosen by an adaptive adversary. This setting extends the classical Linear Quadratic Regulator (LQR) and Linear Quadratic Gaussian (LQG) [22], which assumes known dynamics, full observation, and stochastic Gaussian noise. In contrast, our setting accounts for arbitrary time-varying convex losses, adversarial disturbances, and partial observations. This adversarial setup with partial information falls under the framework of *online nonstochastic control*, see e.g. [19]. We focus on the setting where the system matrices $(A, B, C)$ are known and time-invariant.

---

*Authors ordered alphabetically. Emails: `{ab7728,gon.buzaglo,sd0937,ehazan}@princeton.edu`

39th Conference on Neural Information Processing Systems (NeurIPS 2025).

**Regret Minimization with Partial Observation.** Since minimizing cumulative cost directly is infeasible in adversarial environments, we adopt the standard benchmark of *regret*. The goal is to compete with the best fixed policy $\pi \in \Pi$ in hindsight from a comparator class $\Pi$. Formally, the regret of an algorithm $\mathcal{A}$ is defined as

$$\text{Regret}_T(\mathcal{A}, \Pi) = \sum_{t=1}^{T} c_t(\mathbf{y}_t^{\mathcal{A}}, \mathbf{u}_t^{\mathcal{A}}) - \min_{\pi \in \Pi} \sum_{t=1}^{T} c_t(\mathbf{y}_t^{\pi}, \mathbf{u}_t^{\pi}),$$

where $(\mathbf{y}_t^{\mathcal{A}}, \mathbf{u}_t^{\mathcal{A}})$ are the observation and control sequences induced by the algorithm, and $(\mathbf{y}_t^{\pi}, \mathbf{u}_t^{\pi})$ are those induced by policy $\pi$. Note that since the controller does not observe the full state, the losses have to be defined over observations rather than full states, as regret is meaningful only when costs depend on information available to the controller.

We consider comparator classes $\Pi$ consisting of *linear dynamical controllers* (LDCs), the standard benchmark for optimal control under partial observation [19]. These controllers maintain an internal linear state updated based on incoming observations, and generate control actions via a linear readout of this internal state. As such, LDCs are substantially more expressive than a linear map of the current observation. When the system dynamics and cost functions are known in advance and the disturbances are Gaussian, the optimal controller is computed by the classical *Linear Quadratic Gaussian* (LQG) algorithm [22, 6, 8]. However, in the more general setting that we consider in this work, directly optimizing over LDCs is nonconvex and computationally intractable. To address this, we adopt *improper learning*, using a convex relaxation that enables efficient competition with the best stable LDC in hindsight.

**Marginal Stability and Spectral Filters.** Linear Dynamical Controllers (LDCs) form the most expressive class of linear policies in the online control literature, capturing systems with internal memory and feedback over partial observations [19]. Unlike linear state-feedback or linear action controllers, LDCs can implement rich temporal dependencies and adapt to long-horizon structure—making them the natural comparator class in the partially observed setting.

However, learning or competing with general LDCs is computationally hard without further assumptions. We focus on the regime of *marginal stability*, where the spectral radius of the closed-loop dynamics is at most $1 - \gamma$ for small $\gamma > 0$. This regime is both practically relevant, many systems are designed to retain memory over time, and analytically tractable, as it ensures geometric decay of impulse responses. In practice, small stability margins occur in systems that require smooth, long-memory control such as robotics, thermal regulation, satellite attitude control, surgery, and structural damping. In these applications, aggressive control is infeasible or undesirable, and controllers must operate near the stability boundary to achieve robustness and precision. To make this structure exploitable, we restrict to $(\kappa, \gamma)$-diagonalizably stable LDCs (Definition 3.5), which admit well-conditioned diagonalizations and bounded responses. Appendix D presents an explicit control example where marginal stability yields a $poly(T)$ improvement in cumulative cost.

Our algorithm leverages this structure by expressing the disturbance-response map of LDCs in a spectral basis. We construct a universal set of filters from the top eigenvectors of a Hankel matrix, enabling convex approximation of any such stable controller. This leads to a computationally efficient reduction to online convex optimization, with near-optimal regret and exponential improvement in runtime dependence on the stability margin $\gamma$.

## 1.1 Our Contributions

**New algorithm: Double Spectral Control (DSC).** We propose *Double Spectral Control*, a novel algorithm for controlling partially observed linear dynamical systems (LDSs) with adversarial disturbances and convex losses. DSC constructs a two-level spectral approximation of the best stable linear dynamical controller (LDC): first approximating it by a long-memory open-loop controller, and then expressing that controller as a convolutional operator over the observable signal. This results in a convex parameterization over double-filtered outputs: spectral convolutions of spectral convolutions of past observations.

**Exponential runtime improvement in terms of condition number.** We prove that DSC achieves the regret bound

$$\text{Regret}_T(\text{DSC}) = \mathcal{O}\left(\frac{\sqrt{T}}{\gamma^{11}}\right),$$

where $\gamma$ is the closed-loop stability margin of the best diagonalizably stable LDC. Crucially, the per-step runtime is only $\text{polylog}(T/\gamma)$, exponentially improving over the polynomial dependence of prior work such as GRC [30].

**Empirical Performance.** Preliminary empirical evaluations, presented in Appendix C, provide support for our theoretical findings.

| Method | Regret | Time | Disturbances | Costs |
|---|---|---|---|---|
| LQG | 0 | $O(d^3)$ | i.i.d | Known Quadratic |
| GRC [30] | $\tilde{O}\left(\text{poly}\left(\gamma^{-1}\right)\sqrt{T}\right)$ | $\tilde{O}\left(\text{poly}\left(\gamma^{-1}\right)\log T\right)$ | Adversarial | Online Convex |
| AdaptOn [24] | $\tilde{O}(\text{polylog}(T))$ | $O(d^3 \log T)$ | Stochastic | Strongly Convex |
| **DSC (this work)** | $\tilde{O}\left(\text{poly}\left(\gamma^{-1}\right)\sqrt{T}\right)$ | $\text{polylog}(T/\gamma)$ | Adversarial | Online Convex |

Table 1: Comparison of algorithms for controlling linear dynamical systems under partial observation. Among methods that handle adversarial disturbances and general convex costs, DSC achieves the same asymptotic $\tilde{\mathcal{O}}(\sqrt{T})$ regret as prior work [30], while attaining exponentially faster runtime with respect to the stability margin $\gamma$. For both GRC and DSC, runtime depends only polylogarithmically on the hidden-state dimension $d$.

## 1.2   Related Work

**Control of Linear Dynamical Systems.** Classical control theory provides foundational tools for regulating dynamical systems under uncertainty, with early contributions such as state-space modeling [22] and Lyapunov stability analysis [26]. While optimal control methods like LQR assume full-state observation and stochastic disturbances, modern applications often require adapting to adversarial inputs and partial observability. The online LQG problem corresponds to the setting where the system is driven by well-conditioned, independent Gaussian noise, and the losses $\ell_t(y, u) = y^\top Q y + u^\top R u$ are fixed quadratic functions. LQR is the fully observed analogue of LQG, and the optimal solutions to these problems are known as $\mathcal{H}_2$-optimal controllers, which can be well-approximated by fixed state-feedback controllers (for LQR) or linear dynamical controllers (for LQG).

In the worst-case setting, the $\mathcal{H}_\infty$ framework [7] computes minimax controllers that are optimal against adversarial disturbances and can likewise be represented as LDCs. Extensions such as mixed $\mathcal{H}_2/\mathcal{H}_\infty$ control, risk-sensitive control, and regret-optimal control [33, 16], as well as recent minimax adaptive and output-feedback formulations [23], further explore the trade-off between robustness and performance. In contrast to these worst-case optimal control methods, low-regret algorithms offer instance-wise guarantees that adapt to each realized disturbance sequence. We emphasize that $\mathcal{H}_2$- and $\mathcal{H}_\infty$-optimal controllers for partially observed systems correspond to LDCs, whereas state-feedback controllers suffice only in the fully observed case.

**Online Control and Adversarial Disturbances.** Early work in the machine learning literature on control focused on the online LQR setting [1, 14, 27, 13], establishing $\sqrt{T}$ regret with polynomial runtime. A related line of research [12] investigated online LQR with adversarially chosen quadratic losses, attaining the same $\sqrt{T}$ rate. In all these works, the benchmark is the best linear controller in hindsight. Online control [19] extends the classical setting to environments with unknown and potentially adversarial cost functions and disturbances. The Gradient Perturbation Controller (GPC) [2] achieves sublinear regret against the best linear policy under full state observation, but its runtime scales polynomially with the inverse stability margin. Extensions to strongly convex costs [3] and known quadratic losses [15] have further improved regret bounds in specialized settings. Online control has seen applications in meta-optimization [11], mechanical ventilation [32], and population

regulation [17]. Other recent directions include adaptive control for time-varying systems [29] and online control under bandit feedback [31].

**Partial Observation.** In the setting of fixed quadratic costs and gaussian noise with partial observation of the states, known as the linear–quadratic–Gaussian (LQG) control problem, the optimal solution can be derived using the estimation-control separation principle; see, for example, [8]. [24] study online LQ control with partial observation of the state under stochastic assumptions, which is a more limited setting. In the adversarial case, [30] introduced the Gradient Response Controller (GRC), which parameterizes policies over a latent "nature observation" signal to decouple disturbances from system dynamics. While GRC avoids explicit state estimation and achieves sublinear regret, it retains a polynomial runtime dependence on the stability margin. We propose a new method for this setting, based on a double convolution over universal spectral filters, which achieves the same regret guarantees with exponentially faster runtime.

**Spectral Filtering.** Spectral filtering has been widely used for learning linear dynamical systems from sequences of observations [20]. [21] extended this approach to systems with non-symmetric dynamics, and [28] recently eliminated dimension dependence using Chebyshev approximations. These techniques have been applied primarily in the context of sequence prediction. The only prior use of spectral filtering in control is by [5], who applied it in the offline LQR setting with unknown dynamics.

**Convex Relaxations in Control.** Improper learning has emerged as a powerful tool for bypassing the nonconvexity of optimal control, particularly when competing with rich policy classes. Early work introduced convex surrogates for strongly stable policies [12, 2], and recent advances used spectral filtering to approximate disturbance-response maps in the fully observed setting [9]. Our work develops a new convex relaxation for partial observation, based on a two-level spectral approximation of linear dynamical controllers. Unlike in prediction [20], where convolution with inputs can be directly evaluated using observed outputs, control lacks access to the comparator's actions. In the full observation case, disturbances can be reconstructed and filtered; under partial observation, this is infeasible, so we instead convolve the natural observation sequence (3), introducing additional structure and analytical challenges.

**Online Convex Optimization.** Our approach reduces online control to online convex optimization over spectral policy parameters. For background, we refer to standard references on regret minimization [10, 18].

## 2 Our Method

We propose a convex relaxation of the online control problem with partial observation, grounded in a two-level spectral approximation of linear dynamical controllers (LDCs). The central idea is to represent the input-output behavior of a marginally stable LDC as a composition of spectral operations, enabling efficient improper learning through a compact, universal basis. To do so, we leverage the fact that the controllers in a LDC have the structure of convolution of the natural observations with vectors of the form $[1, \alpha, \alpha^2, \dots]$, and spectral filters provide a compact, universal representation for such sequences. Formally, the filters used in our method are obtained from the top $h$ eigenvectors of a fixed Hankel matrix $H \in \mathbb{R}^{m \times m}$, defined by

$$H_{ij} = \frac{(1 - \gamma)^{i+j-1}}{i + j - 1}, \tag{2}$$

where $\gamma$ is a known lower bound on the system's stability margin. These eigenvectors form a universal basis, independent of the system dynamics, cost functions, or noise realizations, and are precomputed before learning. Figure 1 shows examples of the resulting filter shapes.

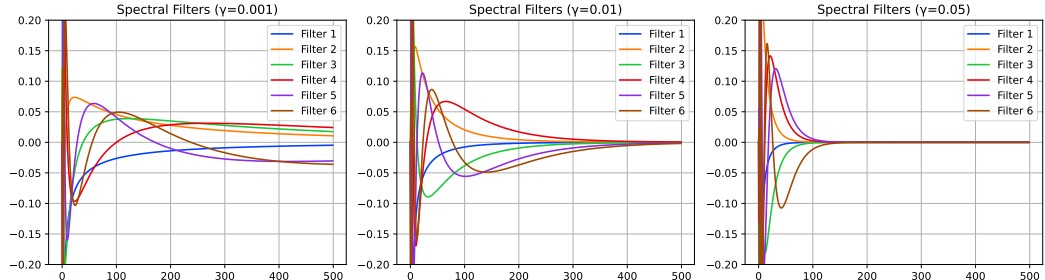

Figure 1: Entries of the first six eigenvectors of $H_{500}$, plotted coordinate-wise.

The algorithm operates on a signal called the *natural observation sequence*, denoted $\mathbf{y}_t^{\mathsf{nat}}$, which corresponds to the output the system would have produced had the learner applied zero controls from the start. This sequence is computed online by maintaining a fictitious internal state $\mathbf{z}_t$ that tracks the contribution of the learner's own actions:

$$\mathbf{z}_{t+1} = A\mathbf{z}_t + B\mathbf{u}_t, \quad \mathbf{y}_t^{\mathsf{nat}} = \mathbf{y}_t - C\mathbf{z}_t. \tag{3}$$

Unlike the raw observations $\mathbf{y}_t$, the natural sequence $\mathbf{y}_t^{\mathsf{nat}}$ is independent of the learner's parameters, making it amenable to convex optimization.

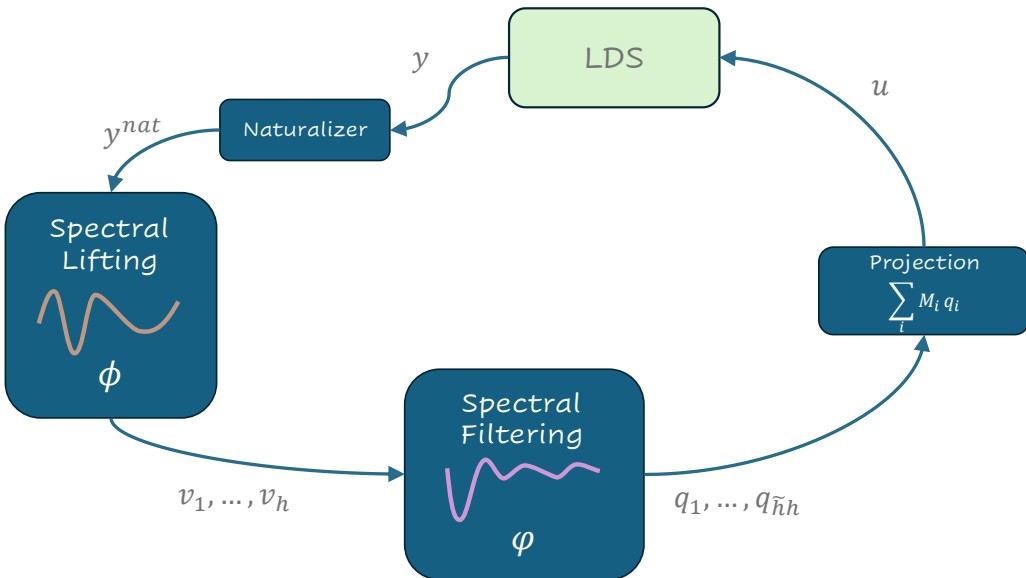

Figure 2: Illustration of the Double Spectral Control (DSC) method. The learner receives the observed signal $\mathbf{y}_t$, from which it computes the natural observation $\mathbf{y}^{\mathsf{nat}}$ in an online manner by maintaining a fictitious internal state. This signal is processed through two levels of spectral operations: a *spectral lifting* stage using filters $\phi$, followed by a *spectral filtering* stage using filters $\varphi$. The resulting features are linearly combined to generate the control action $\mathbf{u}$, which is applied to the linear dynamical system (LDS).

**Why Double Filtering.** As illustrated in 2, our method consists of two stages: *lifting* and *learning*, and spectral filtering is applied in both. If spectral filtering were applied only during learning as in standard lifting approaches such as [19], the ambient dimension would remain polynomial in the stability margin, and the number of tunable parameters would not decrease. While one could, in principle, avoid explicit lifting by analyzing the composite effect of the two filtering operators directly, this would require substantially different assumptions and analysis. We found our double-filtering approach to be both more direct and conceptually aligned with prior work.

**Spectral Filtering vs. Fourier Basis.** While the Fourier basis, used in methods like the Fourier Neural Operator [25], is orthonormal and does not induce spectral decay, the Hankel spectral basis exhibits exponentially decaying eigenvalues. This decay enables logarithmic approximation guarantees in the number of retained components, which are crucial for improved dependence on the stability margin $\gamma$. These guarantees are specific to the Hankel basis and do not directly carry over to Fourier-based representations.

## 2.1 Algorithm Overview

Algorithm 1 constructs the control $\mathbf{u}_t$ through a two-stage spectral transformation of the natural observation history. First, the past sequence $\mathbf{y}_{t-m:t}^{\mathsf{nat}}$ is convolved with a fixed set of spectral filters $\{\phi_j\}_{j=1}^h$, derived from the top eigenvectors of a Hankel matrix $H$. This *spectral lifting* step produces a sequence of intermediate signals that captures long-term dependencies in a compact representation. Crucially, this step enables dimensionality reduction: directly lifting the full history would require a number of parameters that scales polynomially with the inverse stability margin $\gamma^{-1}$, as detailed in previous work [19]. In contrast, our intermediate spectral approximation ensures only polylogarithmic dependence on $\gamma^{-1}$.

Next, the lifted signal undergoes a second spectral transformation, analogous to the spectral filtering technique of [20], as adapted to the control setting by [9], but now applied to the already filtered and lifted natural observation sequence. Specifically, we convolve the lifted signal with a second set of filters $\{\varphi_i\}_{i=1}^{\tilde{h}}$, also obtained as eigenvectors of a Hankel matrix. The resulting features are linearly combined using learnable matrices to produce the control $\mathbf{u}_t$. Since this map is linear in the parameters, we can apply Projected Online Gradient Descent over a convex set $\mathcal{K} \subseteq \mathbb{R}^{(\tilde{h}+1)\times(h+2)\times n\times p}$.

The loss functions $\ell_t$ are convex and memoryless (Definition 3.9). Moreover, by leveraging fast online convolution methods [4], each step can be implemented in time polylogarithmic in the horizon $T$ and the stability margin $\gamma^{-1}$.

---

**Algorithm 1** Double Spectral Control Algorithm

---

1: **Input:** Horizon $T$, number lifting filters $h$, number of learning filters $\tilde{h}$, memories $m, \tilde{m}$, step size $\eta$, convex constrains set $\mathcal{K} \subseteq \mathbb{R}^{(\tilde{h}+1)\times n\times(h+2)p}$.
2: Compute $\{(\sigma_j, \phi_j)\}_{j=1}^h$ and $\{(\lambda_j, \varphi_j)\}_{j=1}^{\tilde{h}}$, the top eigenpairs of a matrices whose $i, j$ th entry is $\frac{(1-\gamma)^{i+j-1}}{i+j-1}$ of dimensions $m$ and $\tilde{m}$ respectively.
3: Initialize $M_i^0 \in \mathbb{R}^{n\times(h+2)p}$ for all $i \in \{0, \dots, \tilde{h}\}$, and $\mathbf{z}_0 = 0 \in \mathbb{R}^d$.
4: **for** $t = 0, \dots, T-1$ **do**
5:    Perform spectral lifting by computing

$$\tilde{\mathbf{y}}_t^{\mathsf{nat}} = \begin{bmatrix} \mathbf{y}_t^{\mathsf{nat}} & \sigma_0^{1/4} Y_{t:t-m}^{\mathsf{nat}} \phi_0 & \cdots & \sigma_h^{1/4} Y_{t:t-m}^{\mathsf{nat}} \phi_h \end{bmatrix}^\top \in \mathbb{R}^{(h+2)p}.$$

6:    Define $\tilde{Y}_{t:t-\tilde{m}}^{\mathsf{nat}} = \begin{bmatrix} \tilde{\mathbf{y}}_t^{\mathsf{nat}} & \cdots & \tilde{\mathbf{y}}_{t-\tilde{m}}^{\mathsf{nat}} \end{bmatrix}$ and compute control

$$\mathbf{u}_t = M_0^t \tilde{\mathbf{y}}_t^{\mathsf{nat}} + \sum_{i=1}^{\tilde{h}} \lambda_i^{1/4} M_i^t \tilde{Y}_{t:t-\tilde{m}}^{\mathsf{nat}} \varphi_i$$

7:    Update $\mathbf{z}_{t+1} = A\mathbf{z}_t + B\mathbf{u}_t$
8:    Observe $\mathbf{y}_{t+1}$ and record $\mathbf{y}_{t+1}^{\mathsf{nat}} = \mathbf{y}_{t+1} - C\mathbf{z}_{t+1}$.
9:    Set $M^{t+1} = \Pi_{\mathcal{K}}\left[M^t - \eta\nabla_M \ell_t\left(M^t\right)\right]$
10: **end for**
11: **return** $M^T$

---

# 3 Preliminaries

## 3.1 Notation

We use $\mathbf{x}$ to denote states, $\mathbf{u}$ for control inputs, $\mathbf{y}$ for observations, and $\mathbf{w}$ for disturbances. The dimensions of the state, control and observation spaces are denoted by $d = \dim(\mathbf{x})$, $n = \dim(\mathbf{u})$ and $p = \dim(\mathbf{y})$ respectively. Matrices related to the system dynamics and control policy are denoted by capital letters $A, B, C, K, M$. For convenience, we write $\mathbf{y}_t^{\mathsf{nat}} = 0$ for all $t \leq 0$.

Given a policy $\pi$, we denote the state and control at time $t$ by $(\mathbf{x}_t^\pi, \mathbf{u}_t^\pi)$ when following $\pi$. If $\pi$ is parameterized by a set of parameters $\Theta$, and the context makes the inputs clear, we use $(\mathbf{x}_t^\Theta, \mathbf{u}_t^\Theta)$ or $(\mathbf{x}_t(\Theta), \mathbf{u}_t(\Theta))$ to refer to the same quantities. For simplicity, we use $(\mathbf{x}_t, \mathbf{u}_t)$ without any superscript or argument to refer to the state and control at time $t$ under Algorithm 1.

## 3.2 Setting

We begin by making the following assumptions about our system. Non-stochasticity allows us to assume without loss of generality that $\mathbf{x}_0 = 0$.

**Definition 3.1.** *An LDS as in* (1) *is controllable if the noiseless LDS given by* $\mathbf{x}_{t+1} = A\mathbf{x}_t + B\mathbf{u}_t$ *can be steered to any target state from any initial state.*

**Assumption 3.2.** *The system matrices $B$ and $C$ are bounded, i.e., $\|B\| \leq \kappa_B, \|C\| \leq \kappa_C$. The powers of the system matrix $A$ are bounded as $\|A^i\| \leq \kappa(1-\gamma)^i$. The disturbance at each time step is also bounded, i.e., $\|\mathbf{w}_t\| \leq W$.*

**Remark 3.3** (Measurement Noise). *Adding bounded noise to the observations is equivalent to introducing an additional bounded disturbance term, which is already captured in our formulation through the definition of $\mathbf{y}^{\mathsf{nat}}$. Due to Assumption 3.2, the same arguments apply when measurement noise is present, and all theoretical guarantees remain valid under this modification.*

**Assumption 3.4.** *The cost functions $c_t(\mathbf{y}, \mathbf{u})$ are convex. Moreover, as long as $\|\mathbf{y}\|, \|\mathbf{u}\| \leq D$, the gradients are bounded:*

$$\|\nabla_{\mathbf{y}} c_t(\mathbf{y}, \mathbf{u})\|, \|\nabla_{\mathbf{u}} c_t(\mathbf{y}, \mathbf{u})\| \leq GD.$$

As mentioned in [19], the most well-known class of controllers for partially observable linear dynamical systems is that of linear dynamical controllers due to its connection to Kalman filtering. We define a $(\kappa, \gamma)-$diagonalizably stable LDCs as follows:

**Definition 3.5.** *A linear dynamical controller $\pi$ has parameters $(A_\pi, B_\pi, C_\pi)$ and chooses the the control at time $t$ as:*

$$\mathbf{s}_{t+1} = A_\pi \mathbf{s}_t + B_\pi \mathbf{y}_t$$
$$\mathbf{u}_t^\pi = C_\pi \mathbf{s}_t$$

*We say that this linear dynamical controller is $(\kappa, \gamma)$-diagonalizably stable if:*

1. *$A_\pi = H_\pi L_\pi H_\pi^{-1}$ where $L_\pi$ is a real positive[2] diagonal matrix such that $\|L_\pi\| \leq 1 - \gamma$ and $\|H_\pi\|, \|H_\pi^{-1}\| \leq \kappa$.*

2. *$\|B_\pi\|, \|C_\pi\| \leq \kappa$.*

3. *Define $\mathcal{A} := \begin{bmatrix} A & BC_\pi \\ B_\pi C & A_\pi \end{bmatrix}$. Then, $\|\mathcal{A}^i\| \leq \kappa^2(1-\gamma)^i$.*

4. *Let $\mathcal{A}_{CL}$ be the closed loop matrix for the policy defined in Lemma A.2, then $\mathcal{A}_{CL} = HLH^{-1}$ where $L$ is a real positive diagonal matrix such that $\|L\| \leq 1 - \gamma$ and $\|H\|, \|H^{-1}\| \leq \kappa$.*

*We denote by $\mathcal{S} = \{(A_\pi, B_\pi, C_\pi) : (A_\pi, B_\pi, C_\pi)$ is $(\kappa, \gamma)$-diagonalizably stable$\}$ the set of such policies, and, with slight abuse of notation, also use $\mathcal{S}$ to refer to the class of LDC policies $\mathbf{u}_t^\pi$ where $(A_\pi, B_\pi, C_\pi) \in \mathcal{S}$.*

---

[2]The requirement of nonnegative eigenvalues can be relaxed by integrating over a larger set; it is imposed here for ease of presentation.

**Assumption 3.6.** *The zero policy* $(A_\pi, B_\pi, C_\pi) = (0, 0, 0)$ *lies in* $\mathcal{S}$.

For simplicity, we assume that $\kappa, \kappa_B, \kappa_C, W \geq 1$ and $\gamma \leq 2/3$, without loss of generality.

Algorithm 1 learns a convex relaxation of the LDC policy class $\mathcal{S}$ from Definition 3.5, which we define as follows:

**Definition 3.7.** *[Double Spectral Controller] The class of Double Spectral Controllers with $h$ lifting filters and $\tilde{h}$ learning filters, memories $(m, \tilde{m})$ and stability $\gamma$ is defined as:*

$$\left\{ \mathbf{u}_t(M) = M_0 \tilde{\mathbf{y}}_t^{\mathsf{nat}} + \sum_{i=1}^{\tilde{h}} \lambda_i^{1/4} M_i \tilde{Y}_{t:t-\tilde{m}}^{\mathsf{nat}} \boldsymbol{\varphi}_i \,\Big|\, M \in \mathbb{R}^{(\tilde{h}+1) \times n \times (h+2)p} \right\} \,,$$

*where:*

1. $\tilde{\mathbf{y}}_t^{\mathsf{nat}} = \begin{bmatrix} \mathbf{y}_t^{\mathsf{nat}} & \sigma_0^{1/4} Y_{t:t-m}^{\mathsf{nat}} \boldsymbol{\phi}_0 & \cdots & \sigma_h^{1/4} Y_{t:t-m}^{\mathsf{nat}} \boldsymbol{\phi}_h \end{bmatrix}^\top \in \mathbb{R}^{(h+2)p}$,

2. $\tilde{Y}_{t:t-\tilde{m}}^{\mathsf{nat}} = \begin{bmatrix} \tilde{\mathbf{y}}_t^{\mathsf{nat}} & \cdots & \tilde{\mathbf{y}}_{t-\tilde{m}}^{\mathsf{nat}} \end{bmatrix}$,

3. $(\sigma_i, \boldsymbol{\phi}_i) \in \mathbb{R} \times \mathbb{R}^m$ and $(\lambda_i, \boldsymbol{\varphi}_i) \in \mathbb{R} \times \mathbb{R}^{\tilde{m}}$ are the $(i+1)$-th and $i$-th top eigenpairs of $H \in \mathbb{R}^{m \times m}$ and $\tilde{H} \in \mathbb{R}^{\tilde{m} \times \tilde{m}}$ with the $(i, j)$-th entry being $\frac{(1-\gamma)^{i+j-1}}{i+j-1}$ respectively.

To enable learning via online gradient descent, we require a bounded set of parameters:

**Definition 3.8.** *We define the domain over which we optimize as*

$$\mathcal{K} = \left\{ M \in \mathbb{R}^{(\tilde{h}+1) \times n \times (h+2)p} \,\Big|\, \left\| \mathbf{y}_t^M \right\|, \left\| \mathbf{u}_t^M \right\| \leq \mathcal{R}, \|M\| \leq \mathcal{R}_M \right\} \,.$$

*where*

$$\mathcal{R} = \frac{4096 \kappa^{24} \kappa_B \kappa_C^2 W h^4}{\gamma^4} \log^{1/2}\left(\frac{2}{\gamma}\right) \,, \quad \mathcal{R}_M = \frac{128 \kappa^{16} \kappa_B \kappa_C \sqrt{h^5 \tilde{h}}}{\gamma^{5/2}} \log^{1/4}\left(\frac{2}{\gamma}\right) \,.$$

We further note that in Algorithm 1, online gradient descent is not performed on the actual cost function, but on a modified cost function, referred to as the memory-less loss function:

**Definition 3.9.** *We define the memory-less loss function at time $t$ as*

$$\ell_t(M) = c_t(\mathbf{y}_t(M), \mathbf{u}_t(M)) \,,$$

*where $\mathbf{u}_t(M)$ is as defined in 3.7, and $\mathbf{y}_t(M)$ is observed by playing $\mathbf{u}_0(M), \ldots, \mathbf{u}_{t-1}(M)$.*

Our use of a memory-less convex loss offers an alternative to [29], which formulates online control as optimization with memory. Our approach, inspired by [2], yields a more direct analysis while still relying on surrogate losses that implicitly capture system memory.

## 4  Main Result and Analysis Overview

In this section, we present our main result and its proof:

**Theorem 4.1** (Main Theorem). *Let $c_t$ be any sequence of convex Lipschitz cost functions satisfying Assumption 3.4, and let the LDS be controllable (Definition 3.1) and satisfy Assumption 3.2. Then, Algorithm 1 achieves the following regret bound:*

$$Regret_T\,(\mathrm{OSC}, \mathcal{S}) = \tilde{\mathcal{O}}\left(\frac{\sqrt{T}}{\gamma^{11}}\right) \,,$$

*where $\tilde{\mathcal{O}}$ hides poly-logarithmic factors in $\frac{T}{\gamma}$ and constants, and $\mathcal{S}$ is the class of LDCs defined in Definition 3.5. This result holds under the following choice of inputs:*

1. $m = \left\lceil \frac{1}{\gamma} \log\left(\frac{C_1 T^{3/2}}{\gamma^3}\right) \right\rceil$,

2. $h = \left\lceil 2 \log T \log \left( \frac{C_2 \sqrt{m}}{\gamma^2} T^{3/2} \log T \log^{1/4} \left( \frac{2}{\gamma} \right) \right) \right\rceil$,

3. $\tilde{m} = \left\lceil \frac{1}{\gamma} \log \left( \frac{C_3 h^{19/2}}{\gamma^{12}} \sqrt{T} \log^{5/4} \left( \frac{2}{\gamma} \right) \right) \right\rceil$,

4. $\tilde{h} = \left\lceil 2 \log T \log \left( \frac{C_4 h^{21/2} \sqrt{\tilde{m}}}{\gamma^{23/2}} \sqrt{T} \log T \log^{3/2} \left( \frac{2}{\gamma} \right) \right) \right\rceil$

5. $\eta = \frac{1}{C_5} \sqrt{\frac{\gamma^7}{h^5 \tilde{h} m \tilde{m}}}$,

6. $K$ is the set from Definition 3.8,

*where the constants are defined as follows:*

$$C_1 = C_0 G \kappa^{13} \kappa_B \kappa_C^4 W^2 , \ C_2 = C_0 G \kappa^{13} \kappa_B^2 \kappa_C^5 W^2 d, \ C_3 = C_0 G \kappa^{56} \kappa_B^3 \kappa_C^5 W^2 ,$$
$$C_4 = C_3 d, \ C_5 = 1024 G \kappa^{12} \kappa_B \kappa_C^3 W^2$$

*and $C_0$ is some absolute constant.*

The regret bound in Theorem 4.1 matches the $\tilde{\mathcal{O}}(\sqrt{T})$ rate that is optimal in the convex cost setting, while significantly improving the computational dependence on the stability margin $\gamma$. Specifically, the number of tunable parameters and the per-round runtime scale only as $\mathrm{polylog}(1/\gamma)$, whereas prior work [30] requires polynomial dependence on $1/\gamma$. Thus, our method preserves the optimal asymptotic regret rate while achieving an exponentially better scaling in both model size and runtime with respect to the stability margin, as emphasized in Corollary 4.4.

Before proceeding to the proof, we outline the key technical definitions and lemmas required to understand the main proof, while deferring their proof to the Appendix.

In Appendix A, we prove that the spectral policy class can approximate $\mathcal{S}$ up to arbitrary accuracy. Formally, we state this as:

**Lemma 4.2.** *For any LDC policy $(A_\pi, B_\pi, C_\pi) \in \mathcal{S}$, there exists a Double spectral controller with $M \in \mathcal{K}$ such that:*

$$\sum_{t=1}^{T} |c_t(\mathbf{y}_t(M), \mathbf{u}(M)) - c_t(\mathbf{y}_t^\pi, \mathbf{u}_t^\pi)| = \mathcal{O}(\sqrt{T}),$$

*for $m, h, \tilde{m}, \tilde{h}$ as defined in Theorem 4.1.*

Classical results in online gradient descent provide a regret bound with respect to loss functions $\ell_t(M^t)$. However, our regret is defined in terms of the actual costs $c_t(\mathbf{x}_t, \mathbf{u}_t)$. To overcome this technicality, in Appendix B.3 we prove that $c_t(\mathbf{x}_t, \mathbf{u}_t)$ is well approximated by $\ell_t(M^t)$. We formally state this result here:

**Lemma 4.3.** *Algorithm 1 is executed with $\eta$ as defined in Theorem 4.1. Then for every $t \in [T]$,*

$$\left| c_t(\mathbf{y}_t, \mathbf{u}_t) - \ell_t(M^t) \right| \leq \frac{16 G \mathcal{R} \mathcal{R}_M h \tilde{h} \sqrt{m \tilde{m}} \kappa^4 \kappa_B \kappa_C^2 W}{\gamma^3 \sqrt{T}} \log^{1/2} \left( \frac{2}{\gamma} \right) .$$

*Proof of Theorem 4.1.* Observe that for our choice of $m, h, \tilde{m}, \tilde{h}$, using Lemma 4.2 and the Definition 3.9 we get:

$$\min_{M^\star \in \mathcal{K}} \sum_{t=1}^{T} \ell_t(M^\star) - \min_{\pi \in \mathcal{S}} \sum_{t=1}^{T} c_t(\mathbf{y}_t^\pi, \mathbf{u}_t^\pi) = \mathcal{O}(\sqrt{T}), \tag{4}$$

since the cost evaluated on the stationary approximating policy $M^\star$ is identical to the memory-less loss on that policy. To derive a regret bound, we apply the standard guarantee for Online Gradient Descent (Theorem 3.1 in [18]). Lemma B.1 ensures that the set $\mathcal{K}$ is convex, and Lemma B.2 confirms the convexity of the memory-less loss functions. Additionally, Lemma B.3 provides a Lipschitz bound for these losses, and together with the diameter bound of $M$, this yields a regret guarantee under our step size choice $\eta$. Instantiating this bound with our selected values of $m, h, \tilde{m}, \tilde{h}$, we obtain:

$$\sum_{t=1}^{T} \ell_t \left(M^t\right) - \min_{M^\star \in \mathcal{K}} \sum_{t=1}^{T} \ell_t \left(M^\star\right) = \tilde{\mathcal{O}} \left( \frac{\sqrt{T}}{\gamma^{11}} \right). \tag{5}$$

Next, Lemma 4.3 gives us the bound on the difference between the memory-less loss $\ell_t(M^t)$ and the cost incurred by the algorithm $c_t(\mathbf{x}_t, \mathbf{u}_t)$ for every $t \in [T]$. Summing over all $t \in [T]$ and for our selected values of $m, h, \tilde{m}, \tilde{h}$:

$$\sum_{t=1}^{T} c_t \left(\mathbf{y}_t, \mathbf{u}_t\right) - \sum_{t=1}^{T} \ell_t \left(M_{1:h}^t\right) = \tilde{\mathcal{O}} \left( \frac{\sqrt{T}}{\gamma^{11}} \right). \tag{6}$$

Finally, we add equations (4),(5),(6) together to obtain the result:

$$\sum_{t=1}^{T} c_t \left(\mathbf{y}_t, \mathbf{u}_t\right) - \min_{\pi \in \mathcal{S}} \sum_{t=1}^{T} c_t \left(\mathbf{y}_t^\pi, \mathbf{u}_t^\pi\right) = \tilde{\mathcal{O}} \left( \frac{\sqrt{T}}{\gamma^{11}} \right)$$

$\square$

Finally, Algorithm 1 maintains only $\text{polylog}\,(T/\gamma)$ parameters at each step $t$. The spectral lifting step (line 5) at time $t \in [T]$ can be efficiently implemented by zero-padding each filter $\phi_i$ to length $T$ and applying online convolution to the natural observation stream $\{\mathbf{y}_t^{\mathsf{nat}}\}_{t \in [T]}$. Likewise, the controller computation (line 6) involves convolving the lifted, filtered stream $\tilde{\mathbf{y}}_t^{\mathsf{nat}}$ with zero-padded filters $\varphi_i$. By leveraging the fast online convolution method presented in Theorem 3 of [4], we obtain the following corollary:

**Corollary 4.4.** *Each round of Algorithm 1 can be implemented with amortized runtime* $\text{polylog}\,(T/\gamma)$.

## 5 Conclusions

We introduced *Double Spectral Control* (DSC), a new algorithm for controlling partially observed linear dynamical systems under adversarial disturbances and convex loss functions. Our method constructs a two-level spectral approximation of linear dynamical controllers, enabling efficient learning via a convex relaxation. DSC achieves optimal regret while exponentially improving the runtime dependence on the stability margin compared to prior work. To our knowledge, this is the first algorithm to combine partial observation, adversarial noise, and general convex losses with polylogarithmic runtime guarantees.

### 5.1 Limitations

Our method assumes access to the system dynamics $(A, B, C)$, which may not be available in many practical scenarios. Extending DSC to settings with unknown or partially known dynamics is an important direction for future work. We also assume full-information feedback on the loss functions; extending the method to the bandit setting is a natural next step.

This work is primarily theoretical and serves as a proof of concept. Our analysis relies on Definition 3.5, which assumes that certain system matrices are real-diagonalizable. While this assumption simplifies the regret analysis, it excludes systems with oscillatory or non-diagonalizable dynamics. The focus on positive real eigenvalues stems from the Hankel-based spectral filtering method rather than an artifact of analysis. Extending the framework to oscillatory or Jordan-block systems remains an open challenge. Empirically, our experiments focused on symmetric systems satisfying these assumptions and therefore should not be interpreted as validating them. Extending the approach to more general dynamical systems, and exploring whether similar spectral compression arises under complex dynamics, presents an important direction for future work. Finally, evaluating DSC on real-world control tasks remains an important step toward validating its practical effectiveness.

## 6 Acknowledgments

EH gratefully acknowledges support from the Office of Naval Research and Open Philanthropy.

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

# A Approximation Results

In this section, we present the proof of Lemma 4.2. We begin by showing that the class $\mathcal{S}$ can be approximated by a family of controllers that are linear in spectral projections of past observations, computed over a fixed memory window. The following definition formalizes this class.

**Definition A.1** (Spectral-Projection Linear Controller). *Let $h \in \mathbb{N}$ denote the number of spectral components, $m \in \mathbb{N}$ the memory length, and $\gamma \in (0, 1)$ a stability parameter. We define the class of* Spectral-Projection Linear Controllers *as:*

$$\left\{ \mathbf{u}_t^K = \sum_{i=1}^{h} \sigma_i^{1/4} K_i Y_{t:t-m}^K \phi_i \;\middle|\; K_i \in \mathbb{R}^{n \times p} \right\},$$

*where $Y_{t:t-m}^K = \begin{bmatrix} y_t^K & y_{t-1}^K & \cdots & y_{t-m}^K \end{bmatrix} \in \mathbb{R}^{p \times (m+1)}$ denotes the matrix of past observations under controller $K$, and $(\sigma_i, \phi_i) \in \mathbb{R} \times \mathbb{R}^{m+1}$ are the top $h + 1$ eigenpairs (indexed from 0) of the Hankel matrix $H \in \mathbb{R}^{(m+1) \times (m+1)}$, with entries*

$$H_{ij} = \frac{(1-\gamma)^{i+j-1}}{i+j-1}.$$

In Section A.1, we establish the following lemma, which shows that the class of linear dynamic controllers in $\mathcal{S}$ can be approximated by the class of spectral-projection linear controllers, for appropriately chosen values of $m$ and $h$.

**Lemma A.2.** *Let a LDC $(A_\pi, B_\pi, C_\pi) \in \mathcal{S}$. Then for*

$$m \geq \frac{1}{\gamma} \log \left( \frac{56 G \kappa^{13} \kappa_B \kappa_C^4 W^2 T}{\varepsilon \gamma^3} \right), \qquad h \geq 2 \log T \log \left( \frac{224 G \kappa^{13} \kappa_B^2 \kappa_C^5 W^2 \sqrt{m} d}{\varepsilon \gamma^2} T \log T \log^{1/4} \left( \frac{2}{\gamma} \right) \right),$$

*and $\varepsilon \in (0, 1)$, there exists a spectral-projection linear controller with*

$$K_i = \sigma_i^{-1/4} \left( \sum_{j=1}^{d} C_\pi H_\pi e_j e_j^\top H_\pi^{-1} B_\pi \phi_i^\top \boldsymbol{\mu}_{\alpha_j} \right) \quad \text{where} \quad \boldsymbol{\mu}_\alpha = [1, \alpha, \dots, \alpha^m] \in \mathbb{R}^{m+1},$$

*such that*

$$\sum_{t=1}^{T} \left| c_t(\mathbf{y}_t^K, \mathbf{u}_t^K) - c_t(\mathbf{y}_t^\pi, \mathbf{u}_t^\pi) \right| \leq \varepsilon T.$$

We now observe that any spectral-projection linear controller can be viewed as a static linear controller acting on a lifted dynamical system. Define the lifted system as follows:

$$\tilde{\mathbf{x}}_t = \tilde{A} \tilde{\mathbf{x}}_{t-1} + \tilde{B} \mathbf{u}_t + \tilde{\mathbf{w}}_t, \qquad \tilde{\mathbf{y}}_t = \tilde{C} \tilde{\mathbf{x}}_t, \tag{7}$$

where

$$\tilde{A} = \begin{bmatrix} A & 0 & \cdots & 0 \\ I & 0 & \cdots & 0 \\ 0 & I & \cdots & 0 \\ \vdots & \vdots & \ddots & \vdots \\ 0 & 0 & \cdots & 0 \end{bmatrix} \in \mathbb{R}^{(m+1)d \times (m+1)d}, \quad \tilde{B} = \begin{bmatrix} B \\ 0 \\ \vdots \\ 0 \end{bmatrix} \in \mathbb{R}^{(m+1)d \times n}, \quad \tilde{\mathbf{w}}_t = \begin{bmatrix} \mathbf{w}_t \\ 0 \\ \vdots \\ 0 \end{bmatrix} \in \mathbb{R}^{(m+1)d}.$$

$$\tag{8}$$

The output matrix $\tilde{C}$ is given by:

$$\tilde{C} = \begin{bmatrix} C & \cdots & 0 \\ \sigma_0^{1/4} \phi_0(1) C & \cdots & \sigma_0^{1/4} \phi_0(m+1) C \\ \vdots & \ddots & \vdots \\ \sigma_h^{1/4} \phi_h(1) C & \cdots & \sigma_h^{1/4} \phi_h(m+1) C \end{bmatrix} \in \mathbb{R}^{(h+2)p \times (m+1)d}, \tag{9}$$

where $\phi_i = (\phi_i(1), \ldots, \phi_i(m+1))^\top$ is the $i+1$-th eigenvector of the Hankel matrix in Definition A.1.

The control law defined by the spectral-projection linear controller corresponds to a static linear controller on the lifted system:

$$K = \begin{bmatrix} 0 \\ K_0 \\ K_1 \\ \vdots \\ K_h \end{bmatrix} \in \mathbb{R}^{(h+2)p \times n}, \quad \text{so that} \quad \mathbf{u}_t = K\tilde{\mathbf{y}}_t. \tag{10}$$

The lifted state and lifted output can be written compactly as:

$$\tilde{\mathbf{x}}_t = \begin{bmatrix} \mathbf{x}_t^\top & \mathbf{x}_{t-1}^\top & \cdots & \mathbf{x}_{t-m}^\top \end{bmatrix}^\top \in \mathbb{R}^{(m+1)d}, \tag{11}$$

$$\tilde{\mathbf{y}}_t^K = \begin{bmatrix} (\mathbf{y}_t^K)^\top & \sigma_0^{1/4}(Y_{t:t-m}^K \phi_0)^\top & \cdots & \sigma_h^{1/4}(Y_{t:t-m}^K \phi_h)^\top \end{bmatrix}^\top \in \mathbb{R}^{(h+2)p}, \tag{12}$$

To evaluate performance, we define the lifted cost $\tilde{c}_t$ by applying the original cost function $c_t$ to the first $p$ coordinates of $\tilde{\mathbf{y}}_t$ (i.e., the original observation $\mathbf{y}_t$) and to the control $\mathbf{u}_t$.

Finally, in the zero-input (natural) system, the lifted observation becomes:

$$\tilde{\mathbf{y}}_t^{\mathsf{nat}} = \begin{bmatrix} (\mathbf{y}_t^{\mathsf{nat}})^\top & \sigma_0^{1/4}(Y_{t:t-m}^{\mathsf{nat}} \phi_0)^\top & \cdots & \sigma_h^{1/4}(Y_{t:t-m}^{\mathsf{nat}} \phi_h)^\top \end{bmatrix}^\top \in \mathbb{R}^{(h+2)p}. \tag{13}$$

where $Y_{t:t-m}^{\mathsf{nat}}$ is as defined in Algorithm 1.

We now summarize key norm bounds for the lifted system components, which will be used in subsequent analysis.

- The output dimension of the lifted system is $(h+2)p$.
- The lifted input matrix satisfies $\|\tilde{B}\| \leq \kappa_B$, and the lifted noise satisfies $\|\tilde{\mathbf{w}}_t\| \leq W$ for all $t$, by assumption.
- By Assumption 3.6, the lifted dynamics matrix satisfies the exponential decay bound
$$\|\tilde{A}^i\| \leq \kappa^2(1-\gamma)^i \quad \text{for all } i \geq 0.$$
- The norm of the lifted output matrix $\tilde{C}$ can be bounded as follows. Each spectral projection row block after the first block in $\tilde{C}$ is of the form
$$\begin{bmatrix} \sigma_i^{1/4}\phi_i(1)C & \cdots & \sigma_i^{1/4}\phi_i(m+1)C \end{bmatrix},$$
and since the eigenvectors $\phi_i \in \mathbb{R}^{m+1}$ are orthonormal and the eigenvalues $\sigma_i$ are bounded by $\sigma_i \leq \log\left(\frac{2}{\gamma}\right)$, defining $\Phi$ as a matrix with $\phi_i^T$ as its rows, we have
$$\|\tilde{C}\| \leq \|C\| + \sigma_{\max}^{1/4}\|\Phi \otimes C\| \leq \kappa_C\left(1 + \sqrt{h+1}\log^{1/4}\left(\frac{2}{\gamma}\right)\right) \leq 4\kappa_C\sqrt{h}\cdot\log^{1/4}\left(\frac{2}{\gamma}\right).$$
- The lifted cost function $\tilde{c}_t$, which is defined by applying the original cost function $c_t$ to the first $p$ coordinates of $\tilde{\mathbf{y}}_t$ and to $\mathbf{u}_t$, inherits the same Lipschitz constant $G$ as the original cost. That is, for all $t$,
$$\text{Lip}(\tilde{c}_t) \leq G.$$
- Finally, for the control matrix $K \in \mathbb{R}^{(h+2)p \times n}$ defined in the proof of Lemma A.2, the norm can be bounded as
$$\|K\| \leq \kappa^4\sqrt{\frac{2(h+1)}{\gamma}} \leq 2\kappa^4\sqrt{\frac{h}{\gamma}}.$$

In section A.2, we prove the following lemma that states that linear controllers can be approximated by spectral controller defined as follows.

**Lemma A.3.** *Consider a system with parameters $(A, B, C)$ such that $\|B\| \leq \tilde{\kappa}_B$, $\|C\| \leq \tilde{\kappa}_C$, and $\|A^i\| \leq \tilde{\kappa}^2(1-\tilde{\gamma})^i$ for all $i \in \mathbb{N}$. Let $\{c_t\}_{t=1}^T$ be a sequence of cost functions that are $\tilde{G}$-Lipschitz over the domain $\{\|\mathbf{y}\|, \|\mathbf{u}\| \leq D\}$. Suppose $K \in \mathbb{R}^{n \times p}$ is a linear controller such that $A + BKC = HLH^{-1}$, where $L$ is a diagonal matrix with nonnegative entries satisfying $\|L\| \leq 1-\tilde{\gamma}$, and $\|H\|, \|H^{-1}\| \leq \tilde{\kappa}$.*

*Then, for any $\varepsilon \in (0, 1)$, there exists a spectral controller of the form*

$$\mathbf{u}_t^M = M_0 \mathbf{y}_t^{\mathsf{nat}} + \sum_{i=1}^{\tilde{h}} \lambda_i^{1/4} M_i Y_{t-1:t-\tilde{m}}^{\mathsf{nat}} \boldsymbol{\varphi}_i,$$

*where $(\lambda_i, \boldsymbol{\varphi}_i)$ are the top $\tilde{h}$ eigenpairs of the Hankel matrix $H \in \mathbb{R}^{\tilde{m} \times \tilde{m}}$ with $H_{ij} = \frac{(1-\tilde{\gamma})^{i+j-1}}{i+j-1}$, such that*

$$\sum_{t=1}^T \left| c_t(\mathbf{y}_t^M, \mathbf{u}_t^M) - c_t(\mathbf{y}_t^K, \mathbf{u}_t^K) \right| \leq 2\varepsilon T,$$

*provided*

$$\tilde{m} \geq \frac{1}{\tilde{\gamma}} \log \left( \frac{6\tilde{G}\tilde{\kappa}^{14}\tilde{\kappa}_B^3 \tilde{\kappa}_C^5 \tilde{W}^2}{\varepsilon \tilde{\gamma}^5} \right), \quad \tilde{h} \geq 2 \log T \cdot \log \left( \frac{400\tilde{G}\tilde{\kappa}^{14}\tilde{\kappa}_B^3 \tilde{\kappa}_C^5 \tilde{W}^2 \sqrt{\tilde{m}} d}{\varepsilon \tilde{\gamma}^{9/2}} \log T \log^{1/4} \left( \frac{2}{\tilde{\gamma}} \right) \right).$$

*Moreover, the parameters $M$ lie in the bounded set*

$$\mathcal{K} = \left\{ M \in \mathbb{R}^{\tilde{h} \times n \times p} \ \middle| \ \|\mathbf{y}_t^M\|, \|\mathbf{u}_t^M\| \leq \frac{4\tilde{\kappa}^6 \tilde{\kappa}_B \tilde{\kappa}_C^2 \tilde{W}}{\tilde{\gamma}^2}, \ \|M_0\| \leq \tilde{\kappa}, \ \|M_i\| \leq \tilde{\kappa}^4 \tilde{\kappa}_B \tilde{\kappa}_C \sqrt{\frac{2}{\tilde{\gamma}}} \right\}.$$

Thus, we complete the proof of Lemma 4.2 as follows:

*Proof of Lemma 4.2.* Substituting the values of the bounds in Lemma A.3 and substituting $\varepsilon = 1/\sqrt{T}$ completes the proof. $\qquad\square$

## A.1 Approximating LDC with spectral Surrogate Controller

In this section, we prove Lemma A.2. We begin by deriving bounds on the observations and controls generated by any linear dynamic controller in $\mathcal{S}$, as well as by the corresponding spectral-projection linear controller constructed in Lemma A.2. These bounds are established in the following lemmas.

**Lemma A.4.** *Let $(A_\pi, B_\pi, C_\pi) \in \mathcal{S}$. Then for all $t \geq 0$, the observation and control satisfy:*

$$\|\mathbf{y}_t^\pi\|, \|\mathbf{u}_t^\pi\| \leq \frac{\kappa_C \kappa^3 W}{\gamma}.$$

*Proof.* Define the joint state $\mathbf{z}_t^\pi = \begin{bmatrix} \mathbf{x}_t^\pi \\ \mathbf{s}_t^\pi \end{bmatrix}$, with closed-loop dynamics

$$\mathbf{z}_{t+1}^\pi = \mathcal{A}\mathbf{z}_t^\pi + \mathcal{B}\mathbf{w}_t, \quad \text{where} \quad \mathcal{A} = \begin{bmatrix} A & BC_\pi \\ B_\pi C & A_\pi \end{bmatrix}, \quad \mathcal{B} = \begin{bmatrix} I \\ 0 \end{bmatrix}.$$

Since $\mathbf{z}_0^\pi = 0$, we have

$$\mathbf{z}_t^\pi = \sum_{i=0}^{t-1} \mathcal{A}^{t-1-i} \mathcal{B}\mathbf{w}_i.$$

The outputs are $\mathbf{y}_t^\pi = [C \ \ 0]\mathbf{z}_t^\pi$, $\mathbf{u}_t^\pi = [0 \ \ C_\pi]\mathbf{z}_t^\pi$. Using definition 3.5, $\|\mathcal{A}^i\| \leq \kappa^2(1-\gamma)^i$, $\|\mathcal{B}\| = 1$, $\|C\| \leq \kappa_C$, $\|C_\pi\| \leq \kappa$, and $\|\mathbf{w}_i\| \leq W$, we get:

$$\|\mathbf{y}_t^\pi\| \leq \kappa_C \sum_{i=0}^{t-1} \|\mathcal{A}^i\| W \leq \frac{\kappa^2 \kappa_C W}{\gamma}, \quad \|\mathbf{u}_t^\pi\| \leq \kappa \sum_{i=0}^{t-1} \|\mathcal{A}^i\| W \leq \frac{\kappa^3 W}{\gamma}.$$

$\qquad\qquad\square$

**Lemma A.5.** *Let $(A_\pi, B_\pi, C_\pi) \in \mathcal{S}$ and let $K$ be as mentioned in Lemma A.2 for $(A_\pi, B_\pi, C_\pi)$. Then, for any $t \geq 0$,*

$$\left\| \mathbf{y}_t^K \right\| \leq \frac{\kappa^2 \kappa_C W}{\gamma}.$$

*Proof.* Observe that our spectral-projection linear controller can be represented by the lifted system defined in Equations (7), (8), (9) and (10) for the choice of $K_i$ in Lemma A.2. By definition 3.5, $\tilde{A} + \tilde{B}\tilde{K}\tilde{C} = HLH^{-1}$, and thus $\left\| (\tilde{A} + \tilde{B}\tilde{K}\tilde{C})^i \right\| \leq \kappa^2 (1 - \gamma)^i$. Moreover, observe that $\|\tilde{\mathbf{w}}_t\| \leq W$. Thus,

$$\tilde{\mathbf{x}}_t^K = \sum_{i=1}^{t} (\tilde{A} + \tilde{B}\tilde{K}\tilde{C})^{i-1} \tilde{\mathbf{w}}_{t-i} \implies \left\| \tilde{\mathbf{x}}_t^K \right\| \leq \frac{\kappa^2 W}{\gamma}.$$

Using Eqn (11), it is clear that $\mathbf{y}_t^K = [C \quad 0 \quad \ldots \quad 0] \tilde{\mathbf{x}}_t^K$ and thus, $\left\| \mathbf{y}_t^K \right\| \leq \frac{\kappa^2 \kappa_C W}{\gamma}$ $\qquad \square$

We now complete the proof of Lemma A.2 as follows:

*Proof of Lemma A.2.* Observe that

$$\mathbf{u}_t^\pi = \sum_{i=0}^{t} C_\pi A_\pi^i B_\pi \mathbf{y}_{t-i}^\pi.$$

Consider the following policy $\pi_\tau$ defined for some $\tau \in \mathbb{N} \cup \{0\}$:

$$\mathbf{u}_t^{\pi_\tau} = \begin{cases} \sum_{i=0}^{h} \sigma_i^{1/4} K_i Y_{t:t-m}^{\pi_\tau} \boldsymbol{\phi}_i & \text{if } t \leq \tau - 1 \\ \sum_{i=0}^{m} C_\pi A_\pi^i B_\pi \mathbf{y}_{t-i}^{\pi_\tau} & \text{if } t = \tau \\ \sum_{i=0}^{t} C_\pi A_\pi^i B_\pi \mathbf{y}_{t-i}^{\pi_\tau} & \text{if } t \geq \tau + 1 \end{cases}$$

Observe that $\pi_0$ is pure LDC policy and that $\pi_{T+1}$ is purely spectral policy. In order to bound the difference of costs between these, we shall bound the difference of costs between $\pi_\tau$ and $\pi_{\tau-1}$ for any $\tau \in [T]$.

Observe that for all $t \in [0, \tau - 2]$, $\mathbf{u}_t^{\pi_\tau} = \mathbf{u}_t^{\pi_{\tau-1}}$ and for all $t \in [0, \tau - 1]$, $\mathbf{y}_t^{\pi_\tau} = \mathbf{y}_t^{\pi_{\tau-1}}$ by the definition of $\pi_\tau$. Thus, $Y_{\tau-1:\tau-m-1}^{\pi_\tau} = Y_{\tau-1:\tau-m-1}^{\pi_{\tau-1}}$ and this gives us

$$\mathbf{u}_{\tau-1}^{\pi_{\tau-1}} = \sum_{i=1}^{m} C_\pi A_\pi^i B_\pi \mathbf{y}_{\tau-i-1}^{\pi_{\tau-1}} = \sum_{i=0}^{m} C_\pi H_\pi L_\pi^i H_\pi^{-1} B_\pi \mathbf{y}_{\tau-i-1}^{\pi_{\tau-1}}.$$

Writing $L_\pi^i = \sum_{j=1}^{d} \alpha_j^i e_j e_j^\top$, and defining $\mu_\alpha = [1 \quad \alpha \quad \ldots \quad \alpha^m]^\top \in \mathbb{R}^{m+1}$, we get

$$\mathbf{u}_{\tau-1}^{\pi_{\tau-1}} = \sum_{i=0}^{m} C_\pi H_\pi \left( \sum_{j=1}^{d} \alpha_j^i e_j e_j^\top \right) H_\pi^{-1} B_\pi \mathbf{y}_{\tau-i-1}^{\pi_{\tau-1}} = \sum_{j=1}^{d} C_\pi H_\pi e_j e_j^\top H_\pi^{-1} B_\pi Y_{\tau-1:\tau-m-1}^{\pi_{\tau-1}} \mu_{\alpha_j}.$$

Since $\{\boldsymbol{\phi}_i \,|\, i \in [m] \cup \{0\}\}$ form an orthonormal basis,

$$\mathbf{u}_{\tau-1}^{\pi_{\tau-1}} = \sum_{j=1}^{d} C_\pi H_\pi e_j e_j^\top H_\pi^{-1} B_\pi Y_{\tau-1:\tau-m-1}^{\pi_{\tau-1}} \left( \sum_{i=0}^{m} \boldsymbol{\phi}_i \boldsymbol{\phi}_i^\top \right) \mu_{\alpha_j} = \sum_{i=0}^{m} \sigma_i^{1/4} K_i Y_{\tau-1:\tau-m-1}^{\pi_{\tau-1}} \boldsymbol{\phi}_i.$$

Since, $Y_{\tau-1:\tau-m-1}^{\pi_{\tau-1}} = Y_{\tau-1:\tau-m-1}^{\pi_\tau}$, and all of $\{\mathbf{y}_t^{\pi_\tau} \,|\, t \in [\tau - 1]\}$ are outputs of the spectral-projection linear policy. Using Lemma A.5, we know that $\left\| Y_{\tau-1:\tau-m-1}^{\pi_\tau} \right\| \leq \frac{\sqrt{m+1}\kappa^2 \kappa_C W}{\gamma} \leq \frac{2\kappa^2 \kappa_C W \sqrt{m}}{\gamma}$. Moreover,

$$\mathbf{u}_{\tau-1}^{\pi_\tau} = \sum_{i=0}^{h} \sigma_i^{1/4} K_i Y_{\tau-1:\tau-m-1}^{\pi_\tau} \boldsymbol{\phi}_i.$$

Taking the difference, we get that:

$$\left\| \mathbf{u}_{\tau-1}^{\pi_\tau} - \mathbf{u}_{\tau-1}^{\pi_{\tau-1}} \right\| \leq \sum_{i=h+1}^{m} \left\| \sigma_i^{1/4} K_i \right\| \left\| Y_{\tau-1:\tau-m-1}^{\pi_\tau} \right\| \leq \frac{2\kappa^6 \kappa_C W \sqrt{m}}{\gamma} \sum_{i=h+1}^{m} \sum_{j=1}^{d} |\boldsymbol{\phi}_i^\top \boldsymbol{\mu}_{\alpha_j}|$$

Using Lemma 7.4 of [9], we get that:

$$\left\|\mathbf{u}_{\tau-1}^{\pi_\tau} - \mathbf{u}_{\tau-1}^{\pi_{\tau-1}}\right\| \leq \frac{2\kappa^6 \kappa_C W d\sqrt{m}}{\gamma} \log^{1/4}\left(\frac{2}{\gamma}\right) \int_h^\infty \exp\left(-\frac{\pi^2 j}{16\log T}\right) dj$$

$$\leq \frac{4\kappa^6 \kappa_C W d\sqrt{m}}{\gamma} \log T \log^{1/4}\left(\frac{2}{\gamma}\right) \exp\left(-\frac{\pi^2 h}{16\log T}\right)$$

$$\leq \delta_1 \qquad \left[h \geq 2\log T \log\left(\frac{4\kappa^6 \kappa_C W d\sqrt{m}}{\delta_1 \gamma} \log T \log^{1/4}\left(\frac{2}{\gamma}\right)\right)\right] \quad (14)$$

Now, observe that since the controls of $\pi_\tau$ and $\pi_{\tau-1}$ are the same until time $\tau - 2$, $\mathbf{x}_{\tau-1}^{\pi_\tau} = \mathbf{x}_{\tau-1}^{\pi_{\tau-1}}$. Thus, using eqn. (14), we get that:

$$\left\|\mathbf{x}_\tau^{\pi_\tau} - \mathbf{x}_\tau^{\pi_{\tau-1}}\right\| = \left\|B(\mathbf{u}_{\tau-1}^{\pi_\tau} - \mathbf{u}_{\tau-1}^{\pi_{\tau-1}})\right\| \leq \kappa_B \delta_1 \quad (15)$$

and

$$\left\|\mathbf{y}_\tau^{\pi_\tau} - \mathbf{y}_\tau^{\pi_{\tau-1}}\right\| = \left\|CB(\mathbf{u}_{\tau-1}^{\pi_\tau} - \mathbf{u}_{\tau-1}^{\pi_{\tau-1}})\right\| \leq \kappa_B \kappa_C \delta_1 \quad (16)$$

Again by using the fact that $\mathbf{y}_{t-i}^{\pi_\tau} = \mathbf{y}_{t-i}^{\pi_{\tau-1}}$ for all $i \geq 1$,

$$\left\|\mathbf{u}_\tau^{\pi_\tau} - \mathbf{u}_\tau^{\pi_{\tau-1}}\right\| = \left\|\sum_{i=0}^\tau C_\pi A_\pi^i B_\pi \mathbf{y}_{\tau-i}^{\pi_\tau} - \sum_{i=0}^m C_\pi A_\pi^i B_\pi \mathbf{y}_{\tau-i}^{\pi_{\tau-1}}\right\|$$

$$\leq \left\|C_\pi B_\pi \left(\mathbf{y}_\tau^{\pi_\tau} - \mathbf{y}_\tau^{\pi_{\tau-1}}\right)\right\| + \left\|\sum_{i=1}^\tau C_\pi A_\pi^i B_\pi \mathbf{y}_{\tau-i}^{\pi_\tau} - \sum_{i=1}^m C_\pi A_\pi^i B_\pi \mathbf{y}_{\tau-i}^{\pi_{\tau-1}}\right\|$$

$$\leq \kappa_B \kappa_C \kappa^2 \delta_1 + \frac{\kappa^6 \kappa_C W}{\gamma^2} (1-\gamma)^{m+1} \qquad \text{[choice of } h]$$

$$\leq 2\kappa^2 \kappa_B \kappa_C \delta_1 \qquad \left[m \geq \frac{1}{\gamma} \log\left(\frac{\kappa^4 W}{\kappa_B \gamma^2 \delta_1}\right)\right] \quad (17)$$

Now, for any $t > \tau$, we have that

$$\begin{bmatrix} \mathbf{x}_t^{\pi_\tau} - \mathbf{x}_t^{\pi_{\tau-1}} \\ \mathbf{s}_t^{\pi_\tau} - \mathbf{s}_t^{\pi_{\tau-1}} \end{bmatrix} = \begin{bmatrix} A & B_\pi C \\ BC_\pi & A_\pi \end{bmatrix} \begin{bmatrix} \mathbf{x}_{t-1}^{\pi_\tau} - \mathbf{x}_{t-1}^{\pi_{\tau-1}} \\ \mathbf{s}_{t-1}^{\pi_\tau} - \mathbf{s}_{t-1}^{\pi_{\tau-1}} \end{bmatrix} = \begin{bmatrix} A & B_\pi C \\ BC_\pi & A_\pi \end{bmatrix}^{t-\tau-1} \begin{bmatrix} \mathbf{x}_{\tau+1}^{\pi_\tau} - \mathbf{x}_{\tau+1}^{\pi_{\tau-1}} \\ \mathbf{s}_{\tau+1}^{\pi_\tau} - \mathbf{s}_{\tau+1}^{\pi_{\tau-1}} \end{bmatrix}.$$

In order to compute the terms on the RHS,

$$\left\|\mathbf{x}_{\tau+1}^{\pi_\tau} - \mathbf{x}_{\tau+1}^{\pi_{\tau-1}}\right\| = \left\|A\left(\mathbf{x}_\tau^{\pi_\tau} - \mathbf{x}_\tau^{\pi_{\tau-1}}\right) + B\left(\mathbf{u}_\tau^{\pi_\tau} - \mathbf{u}_\tau^{\pi_{\tau-1}}\right)\right\|$$

$$\leq \kappa^2 \left\|\mathbf{x}_\tau^{\pi_\tau} - \mathbf{x}_\tau^{\pi_{\tau-1}}\right\| + \kappa_B \left\|\mathbf{u}_\tau^{\pi_\tau} - \mathbf{u}_\tau^{\pi_{\tau-1}}\right\|$$

$$\leq \kappa^2 \kappa_B \delta_1 + 2\kappa^2 \kappa_B^2 \kappa_C \delta_1 \leq 3\kappa^2 \kappa_B^2 \kappa_C \delta_1. \qquad \text{[Eqns (15) and (17)]} \quad (18)$$

and thus,

$$\left\|\mathbf{y}_{\tau+1}^{\pi_\tau} - \mathbf{y}_{\tau+1}^{\pi_{\tau-1}}\right\| = \left\|C(\mathbf{x}_{\tau+1}^{\pi_\tau} - \mathbf{x}_{\tau+1}^{\pi_{\tau-1}})\right\| \leq 3\kappa^2 \kappa_B^2 \kappa_C^2 \delta_1 \quad (19)$$

Moreover, for $t > \tau$, the controller does behave like an LDC, however, the starting state is different and must be calculated. Observe that if $\mathbf{s}_t^{\pi_\tau} := \sum_{i=0}^t A_\pi^i B_\pi \mathbf{y}_{t-i}^{\pi_\tau}$ for all $t > \tau$, then the control is correctly defined. Hence,

$$\left\|\mathbf{s}_{\tau+1}^{\pi_\tau} - \mathbf{s}_{\tau+1}^{\pi_{\tau-1}}\right\| = \left\|A_\pi B_\pi \left(\mathbf{y}_\tau^{\pi_\tau} - \mathbf{y}_\tau^{\pi_{\tau-1}}\right) + B_\pi \left(\mathbf{y}_{\tau+1}^{\pi_\tau} - \mathbf{y}_{\tau+1}^{\pi_{\tau-1}}\right)\right\|$$

$$\leq \kappa^3 \kappa_B \kappa_C \delta_1 + 3\kappa^3 \kappa_B^2 \kappa_C^2 \delta_1 \leq 4\kappa^3 \kappa_B^2 \kappa_C^2 \delta_1 \qquad \text{[Eqns (16) and (19)]}.$$

Putting this together, we have that

$$\left\|\begin{bmatrix} \mathbf{x}_{\tau+1}^{\pi_\tau} - \mathbf{x}_{\tau+1}^{\pi_{\tau-1}} \\ \mathbf{s}_{\tau+1}^{\pi_\tau} - \mathbf{s}_{\tau+1}^{\pi_{\tau-1}} \end{bmatrix}\right\| \leq 3\kappa^2 \kappa_B^2 \kappa_C \delta_1 + 4\kappa^3 \kappa_B^2 \kappa_C^2 \delta_1 \leq 7\kappa^3 \kappa_B^2 \kappa_C^2 \delta_1.$$

Thus, we have

$$\left\|\begin{bmatrix} \mathbf{x}_t^{\pi_\tau} - \mathbf{x}_t^{\pi_{\tau-1}} \\ \mathbf{s}_t^{\pi_\tau} - \mathbf{s}_t^{\pi_{\tau-1}} \end{bmatrix}\right\| \leq \kappa^2 (1-\gamma)^{t-\tau-1} \left\|\begin{bmatrix} \mathbf{x}_{\tau+1}^{\pi_\tau} - \mathbf{x}_{\tau+1}^{\pi_{\tau-1}} \\ \mathbf{s}_{\tau+1}^{\pi_\tau} - \mathbf{s}_{\tau+1}^{\pi_{\tau-1}} \end{bmatrix}\right\| \leq 7\kappa^5 \kappa_B^2 \kappa_C^2 \delta_1.$$

Observe that

$$\mathbf{y}_t^{\pi_\tau} - \mathbf{y}_t^{\pi_{\tau-1}} = [C \quad 0]\begin{bmatrix} \mathbf{x}_t^{\pi_\tau} - \mathbf{x}_t^{\pi_{\tau-1}} \\ \mathbf{s}_t^{\pi_\tau} - \mathbf{s}_t^{\pi_{\tau-1}} \end{bmatrix}, \qquad \mathbf{y}_t^{\pi_\tau} - \mathbf{y}_t^{\pi_{\tau-1}} = [0 \quad C_\pi]\begin{bmatrix} \mathbf{x}_t^{\pi_\tau} - \mathbf{x}_t^{\pi_{\tau-1}} \\ \mathbf{s}_t^{\pi_\tau} - \mathbf{s}_t^{\pi_{\tau-1}} \end{bmatrix},$$

for all $t > \tau$. Thus, for all $t > \tau$ (and also for all $t \in [T]$ using the previous bounds) we have that

$$\max\left\{\left\|\mathbf{y}_t^{\pi_\tau} - \mathbf{y}_t^{\pi_{\tau-1}}\right\|, \left\|\mathbf{u}_t^{\pi_\tau} - \mathbf{u}_t^{\pi_{\tau-1}}\right\|\right\} \le 7\kappa^6\kappa_B^2\kappa_C^3\delta_1.$$

Now, observe that:

$$\max\left\{\left\|\mathbf{y}_t^{\pi_{T+1}} - \mathbf{y}_t^{\pi_0}\right\|, \left\|\mathbf{u}_t^{\pi_{T+1}} - \mathbf{u}_t^{\pi_0}\right\|\right\} \le \sum_{\tau=1}^{T+1}\max\left\{\left\|\mathbf{y}_t^{\pi_\tau} - \mathbf{y}_t^{\pi_{\tau-1}}\right\|, \left\|\mathbf{u}_t^{\pi_\tau} - \mathbf{u}_t^{\pi_{\tau-1}}\right\|\right\}$$

$$\le 14\kappa^6\kappa_B^2\kappa_C^3\delta_1 T$$

If out choice of $\delta_1 \le W/14\kappa^3\kappa_B^2\kappa_C^2\gamma T$, then the difference is upper bounded by $\kappa^3\kappa_C W/\gamma$ and since by lemma A.4, $\|\mathbf{y}_t^{\pi_0}\|, \|\mathbf{u}_t^{\pi_0}\| \le \kappa^3\kappa_C W/\gamma$, we have that $\|\mathbf{y}_t^{\pi_0}\|, \|\mathbf{u}_t^{\pi_0}\|, \|\mathbf{y}_t^{\pi_{T+1}}\|, \|\mathbf{u}_t^{\pi_{T+1}}\| \le 2\kappa^3\kappa_C W/\gamma$. Using lipschitzness, this gives us

$$\sum_{t=1}^T \left|c_t(\mathbf{y}_t^{\pi_{T+1}}, \mathbf{u}_t^{\pi_{T+1}}) - c_t(\mathbf{y}_t^{\pi_0}, \mathbf{u}_t^{\pi_0})\right| \le \frac{2G\kappa^3\kappa_C W}{\gamma}\left(\sum_{t=1}^T \left\|\mathbf{y}_t^{\pi_{T+1}} - \mathbf{y}_t^{\pi_0}\right\| + \left\|\mathbf{u}_t^{\pi_{T+1}} - \mathbf{u}_t^{\pi_0}\right\|\right)$$

$$\le \frac{2G\kappa^3\kappa_C W}{\gamma} \cdot T \cdot 28\kappa^6\kappa_B^2\kappa_C^3\delta_1 T$$

$$\le \frac{56G\kappa^9\kappa_B^2\kappa_C^4 W}{\gamma} \cdot \delta_1 T^2 = \varepsilon T$$

by picking $\delta_1 = \varepsilon\gamma/56G\kappa^9\kappa_B^2\kappa_C^4 WT \le W/14\kappa^3\kappa_B^2\kappa_C^2\gamma T$. This completes the proof by substituting the value of $\delta_1$ into the choosen expressions of $m$ and $h$.

$\square$

## A.2 Linear controllers

In the previous section, we have proven that any LDC controller in $\tilde{\mathcal{S}}$ can be approximated by spectral-projection linear controller. As explained previously, this spectral-projection linear controller can be viewed as a linear controller over the lifted system defined in eqns (7), (8), (9).

In this section, we prove that the class of linear controllers can be approximated by spectral filtering. We begin by making certain assumptions only for this section, and defining the class of linear controllers that we aim to compete against and the class of spectral controllers.

**Assumption A.6.** *The system matrices $B, C$ are bounded, i.e., $\|B\| \le \tilde{\kappa}_B, \|C\| \le \tilde{\kappa}_C$. The disturbance at each time step is also bounded, i.e., $\|\mathbf{w}_t\| \le \tilde{W}$.*

**Assumption A.7.** *The cost functions $c_t(\mathbf{y}, \mathbf{u})$ are convex. Moreover, as long as $\|\mathbf{y}\|, \|\mathbf{u}\| \le D$, the gradients are bounded:*
$$\|\nabla_\mathbf{y} c_t(\mathbf{y}, \mathbf{u})\|, \|\nabla_\mathbf{u} c_t(\mathbf{y}, \mathbf{u})\| \le \tilde{G}D.$$

**Definition A.8.** *A linear policy $K$ is $(\tilde{\kappa}, \tilde{\gamma})$-diagonalizably stable if there exist matrices $L, H$ satisfying $A + BKC = HLH^{-1}$, such that the following conditions hold:*

1. *$L$ is diagonal with nonnegative entries.*

2. *The spectral norm of $L$ is strictly less than one, i.e., $\|L\| \le 1 - \tilde{\gamma}$.*

3. *The controller and the transformation matrices are bounded, i.e., $\|K\|, \|H\|, \|H^{-1}\| \le \tilde{\kappa}$.*

*We denote by $\tilde{\mathcal{S}} = \{K : K \text{ is } (\tilde{\kappa}, \tilde{\gamma})\text{-diagonalizably stable}\}$ the set of such policies, and, with slight abuse of notation, also use $\tilde{\mathcal{S}}$ to refer to the class of linear policies $\mathbf{u}_t = S\mathbf{y}_t$ where $S \in \tilde{\mathcal{S}}$. Each policy in $\tilde{\mathcal{S}}$ is fully parameterized by the matrix $K \in \mathbb{R}^{n \times p}$.*

**Assumption A.9.** *The zero policy $K = 0$ lies in $\tilde{S}$.*

For simplicity, we assume that $\tilde{\kappa}, \tilde{\kappa}_B, \tilde{W}, \tilde{G} \geq 1$ and $\tilde{\gamma} \leq 2/3$, without loss of generality.

We define $\mathbf{y}_t^{\mathsf{nat}}$ as the observation at time $t$ assuming that all inputs to the system were zero from the beginning of time, i.e.

$$\mathbf{x}_{t+1}^{\mathsf{nat}} = A\mathbf{x}_t^{\mathsf{nat}} + \mathbf{w}_t; \qquad \mathbf{y}_t^{\mathsf{nat}} = C\mathbf{x}_t^{\mathsf{nat}}$$

We define our disturbance response controller with respect to these as follows:

**Definition A.10.** *[Spectral Controller] The class of Spectral Controllers with $h$ parameters, memory $m$ and stability $\tilde{\gamma}$ is defined as:*

$$\left\{ \mathbf{u}_t^M = M_0 \mathbf{y}_t^{\mathsf{nat}} + \sum_{i=1}^h \lambda_i^{1/4} M_i Y_{t-1:t-m}^{\mathsf{nat}} \boldsymbol{\varphi}_i \right\},$$

*where $\boldsymbol{\varphi}_i \in \mathbb{R}^m, \lambda_i \in \mathbb{R}$ are the $i^{\mathsf{th}}$ top eigenvector and eigenvalue of $H \in \mathbb{R}^{m \times m}$ such that $H_{ij} = \frac{(1-\tilde{\gamma})^{i+j-1}}{i+j-1}$. Any policy in this class is fully parameterized by the matrices $M \in \mathbb{R}^{n \times p \times h}$.*

The sequence $\{\mathbf{y}_t^{\mathsf{nat}}\}_{t \in [T]}$ can be iteratively computed using the following:

$$\mathbf{y}_t^{\mathsf{nat}} = \mathbf{y}_t - C \sum_{i=1}^t A^{i-1} B \mathbf{u}_{t-i}$$

Alternatively, it can be carried out recursively, starting from $\mathbf{z}_0 = 0$, as:

$$\mathbf{z}_{t+1} = A\mathbf{z}_t + B\mathbf{u}_t; \qquad \mathbf{y}_t^{\mathsf{nat}} = \mathbf{y}_t - C\mathbf{z}_t$$

**Lemma A.11.** *Under our assumptions, $\|\mathbf{y}_t^{\mathsf{nat}}\| \leq \frac{\tilde{\kappa}^2 \tilde{\kappa}_C \tilde{W}}{\tilde{\gamma}}$*

*Proof.* Unrolling the recursion, we get that

$$\mathbf{y}_t^{\mathsf{nat}} = \sum_{i=1}^t C A^{i-1} \mathbf{w}_{t-i}$$

Taking norm on both sides,

$$\left\|\mathbf{y}_t^{\mathsf{nat}}\right\| \leq \tilde{\kappa}^2 \tilde{\kappa}_C \tilde{W} \sum_{i=1}^t (1-\tilde{\gamma})^{i-1} \leq \frac{\tilde{\kappa}^2 \tilde{\kappa}_C \tilde{W}}{\tilde{\gamma}}$$

$\square$

**Lemma A.12.** *For any $K \in \tilde{S}$, the corresponding states $\mathbf{x}_t^K$ and control inputs $\mathbf{u}_t^K$ are bounded by*

$$\left\|\mathbf{y}_t^K\right\| \leq \frac{2\tilde{\kappa}^5 \tilde{\kappa}_B \tilde{\kappa}_C^2 \tilde{W}}{\tilde{\gamma}^2}, \quad \left\|\mathbf{u}_t^K\right\| \leq \frac{2\tilde{\kappa}^6 \tilde{\kappa}_B \tilde{\kappa}_C^2 \tilde{W}}{\tilde{\gamma}^2}.$$

*Proof.* As in many other parts of this paper, we first write $\Delta_t^K$ as a linear transformation of nature's observations:

$$\left\|\Delta_t^K\right\| = \left\|\sum_{i=1}^t (A + BKC)^{i-1} BK \mathbf{y}_{t-i}^{\mathsf{nat}}\right\|$$

$$\leq \tilde{\kappa}^3 \tilde{\kappa}_B \sum_{i=0}^t (1-\tilde{\gamma})^i \left\|\mathbf{y}_{t-i-1}^{\mathsf{nat}}\right\|$$

$$\leq \frac{\tilde{\kappa}^5 \tilde{\kappa}_B \tilde{\kappa}_C \tilde{W}}{\tilde{\gamma}^2}.$$

Then, since $\mathbf{y}_t^K = \mathbf{y}_t^{\mathsf{nat}} + C\Delta_t^K$ and $\mathbf{u}_t^K = K\mathbf{y}_t^K$, we get our result. $\square$

For convenience of notation, for a given policy $\pi$ we define $\Delta_t^\pi = \mathbf{x}_t^\pi - \mathbf{x}_t^{\text{nat}}$. We begin by defining the class of open-loop optimal controllers as follows:

**Definition A.13** (Open Loop Optimal Controller). *The class of Open Loop Optimal Controllers of with memory $m$ is defined as:*

$$\left\{ \mathbf{u}_t^{K,m} = K\mathbf{y}_t^{\text{nat}} + \sum_{i=1}^m KC\left(A + BKC\right)^{i-1} BK\mathbf{y}_{t-i}^{\text{nat}} \right\} .$$

*Any policy in this class is fully parameterized by the matrix $K \in \mathbb{R}^{d \times n}$ and the memory $m \in \mathbb{Z}$.*

Next, we state and prove Lemma A.14, which shows that any linear policy in $\tilde{\mathcal{S}}$ can be approximated up to arbitrary accuracy with an open-loop optimal controller of suitable memory.

**Lemma A.14.** *Let a linear policy $K \in \tilde{\mathcal{S}}$. Then, for $m \geq \frac{1}{\tilde{\gamma}} \log\left(\frac{6\tilde{G}\tilde{\kappa}^{14}\tilde{\kappa}_B^3\tilde{\kappa}_C^5\tilde{W}^2}{\varepsilon\tilde{\gamma}^5}\right)$ and $\varepsilon \in (0,1)$,*

$$\sum_{t=1}^T \left| c_t(\mathbf{y}_t^{K,m}, \mathbf{u}_t^{K,m}) - c_t(\mathbf{y}_t^K, \mathbf{u}_t^K) \right| \leq \varepsilon T, \qquad \left\| \mathbf{y}_t^{K,m} \right\|, \left\| \mathbf{u}_t^{K,m} \right\| \leq \frac{3\tilde{\kappa}^6\tilde{\kappa}_B\tilde{\kappa}_C^2\tilde{W}}{\tilde{\gamma}^2} .$$

*Proof.* We begin by bounding the difference in the observation and the difference in the control inputs. The difference in cost is bounded using the fact that the cost functions are lipschitz in the control and observation. We begin by unrolling the expressions for $\Delta_t^K$ in terms of $\mathbf{y}_t^{\text{nat}}$:

$$\Delta_t^K = A\Delta_{t-1}^K + BK\mathbf{y}_{t-1}^K = (A + BKC)\Delta_{t-1}^K + BK\mathbf{y}_{t-1}^{\text{nat}} = \sum_{i=1}^t (A + BKC)^{i-1} BK\mathbf{y}_{t-i}^{\text{nat}}$$

$$(20)$$

This gives us that:

$$\mathbf{u}_t^K = K\mathbf{y}_t^{\text{nat}} + KC\Delta_t^K = K\mathbf{y}_t^{\text{nat}} + \sum_{i=1}^t KC(A + BKC)^{i-1} BK\mathbf{y}_{t-i}^{\text{nat}}$$

Notice that for all $t \leq m$, $\mathbf{u}_t^K = \mathbf{u}_t^{K,m}$ and hence $\mathbf{y}_t^K = \mathbf{y}_t^{K,m}$. For $t > m$,

$$\begin{aligned}
\left\| \mathbf{u}_t^K - \mathbf{u}_t^{K,m} \right\| &= \left\| \sum_{i=m+1}^t KC(A + BKC)^{i-1} BK\mathbf{y}_{t-i}^{\text{nat}} \right\| \\
&\leq \frac{\tilde{\kappa}^6\tilde{\kappa}_B\tilde{\kappa}_C^2\tilde{W}}{\tilde{\gamma}} \sum_{i=m+1}^t (1-\tilde{\gamma})^{i-1} \qquad [\Delta - \text{ineq., C-S}] \\
&\leq \frac{\tilde{\kappa}^6\tilde{\kappa}_B\tilde{\kappa}_C^2\tilde{W}}{\tilde{\gamma}^2} (1-\tilde{\gamma})^m .
\end{aligned}$$

We also note that for any policy $\pi$,

$$\mathbf{y}_t^\pi = \mathbf{y}_t^{\text{nat}} + C\Delta_t^\pi = \mathbf{y}_t^{\text{nat}} + \sum_{i=1}^t CA^{i-1} B\mathbf{u}_{t-i}^\pi$$

Using both the previous results and the fact that $\mathbf{u}_t^K = \mathbf{u}_t^{K,m}$ for any $t \leq m$, we similarly get:

$$\left\| \mathbf{y}_t^K - \mathbf{y}_t^{K,m} \right\| = \left\| \sum_{i=1}^t CA^{i-1} B(\mathbf{u}_{t-i}^K - \mathbf{u}_{t-i}^{K,m}) \right\| = \left\| \sum_{i=1}^{t-m} CA^{i-1} B(\mathbf{u}_{t-i}^K - \mathbf{u}_{t-i}^{K,m}) \right\| ,$$

and by using Assumption A.9 we can write:

$$\left\| \mathbf{y}_t^K - \mathbf{y}_t^{K,m} \right\| \leq \tilde{\kappa}_B\tilde{\kappa}_C\tilde{\kappa}^2 \sum_{i=1}^\infty (1-\tilde{\gamma})^{i-1} \left\| \mathbf{u}_{t-i}^K - \mathbf{u}_{t-i}^{K,m} \right\| \leq \frac{\tilde{\kappa}^8\tilde{\kappa}_B^2\tilde{\kappa}_C^3\tilde{W}}{\tilde{\gamma}^3} (1-\tilde{\gamma})^m .$$

Using the fact that $\tilde{\kappa}_B \tilde{\kappa}_C \tilde{\kappa}^2 / \tilde{\gamma} > 1$, we get the following uniform bound:

$$\max \left\{ \left\| \mathbf{u}_t^K - \mathbf{u}_t^{K,m} \right\|, \left\| \mathbf{y}_t^K - \mathbf{y}_t^{K,m} \right\| \right\} \le \frac{\tilde{\kappa}^8 \tilde{\kappa}_B^2 \tilde{\kappa}_C^3 \tilde{W}}{\tilde{\gamma}^3} (1 - \tilde{\gamma})^m .$$

Whenever $m \ge \frac{1}{\tilde{\gamma}} \log \left( \frac{\tilde{\kappa}^2 \tilde{\kappa}_B \tilde{\kappa}_C}{\tilde{\gamma}} \right)$, which is indeed true from our choice of $\varepsilon$ and $m$, this implies that the $\left\| \mathbf{y}_t^K - \mathbf{y}_t^{K,m} \right\|, \left\| \mathbf{u}_t^K - \mathbf{u}_t^{K,m} \right\| \le \tilde{\kappa}^6 \tilde{\kappa}_B \tilde{\kappa}_C^2 \tilde{W} / \tilde{\gamma}^2$. Using Lemma A.12 $\left\| \mathbf{y}_t^K \right\|, \left\| \mathbf{u}_t^K \right\| \le 2 \tilde{\kappa}^6 \tilde{\kappa}_B \tilde{\kappa}_C^2 \tilde{W} / \tilde{\gamma}^2$, by triangle inequality $\left\| \mathbf{y}_t^{K,m} \right\|, \left\| \mathbf{u}_t^{K,m} \right\| \le 3 \tilde{\kappa}^6 \tilde{\kappa}_B \tilde{\kappa}_C^2 \tilde{W} / \tilde{\gamma}^2$. Thus, the sum of costs is bounded using lipschitzness of $c_t$ as follows:

$$\sum_{t=1}^{T} \left| c_t(\mathbf{y}_t^K, \mathbf{u}_t^K) - c_t(\mathbf{y}_t^{K,m}, \mathbf{u}_t^{K,m}) \right| \le \frac{3 \tilde{G} \tilde{\kappa}^6 \tilde{\kappa}_B \tilde{\kappa}_C^2 \tilde{W}}{\tilde{\gamma}^2} \sum_{t=1}^{T} \left( \left\| \mathbf{y}_t^K - \mathbf{y}_t^{K,m} \right\| + \left\| \mathbf{u}_t^K - \mathbf{u}_t^{K,m} \right\| \right)$$

$$\le \frac{6 \tilde{G} \tilde{\kappa}^{14} \tilde{\kappa}_B^3 \tilde{\kappa}_C^5 \tilde{W}^2}{\tilde{\gamma}^5} (1 - \tilde{\gamma})^m \cdot T \le \varepsilon T . \qquad \text{[choice of } m\text{]}$$

$\square$

To enable learning via online gradient descent, we require a bounded set of parameters:

**Definition A.15.** *The set of bounded spectral parameters is defined as*

$$\mathcal{K} = \left\{ M \in \mathbb{R}^{h \times n \times p} \mid \left\| \mathbf{y}_t^M \right\|, \left\| \mathbf{u}_t^M \right\| \le \frac{4 \tilde{\kappa}^6 \tilde{\kappa}_B \tilde{\kappa}_C^2 \tilde{W}}{\tilde{\gamma}^2}, \|M_0\| \le \tilde{\kappa}, \|M_i\| \le \tilde{\kappa}^4 \tilde{\kappa}_B \tilde{\kappa}_C \sqrt{\frac{2}{\tilde{\gamma}}} \right\} .$$

We shall now prove that every open-loop optimal controller can be approximated up to arbitrary accuracy with a spectral controller.

**Lemma A.16.** *For every open loop optimal controller $\pi_{K,m}^{\mathsf{OLOC}}$ such that $K \in \tilde{\mathcal{S}}$ and $\left\| \mathbf{y}^{K,m} \right\|, \left\| \mathbf{u}^{K,m} \right\| \le \frac{3 \tilde{\kappa}^6 \tilde{\kappa}_B \tilde{\kappa}_C^2 \tilde{W}}{\tilde{\gamma}^2}$, there exists an spectral controller $\pi_{h,m,\tilde{\gamma},M}^{\mathsf{SC}}$ with $M \in \mathcal{K}$ such that:*

$$\sum_{t=1}^{T} \left| c_t(\mathbf{y}_t^M, \mathbf{u}_t^M) - c_t(\mathbf{y}_t^{K,m}, \mathbf{u}_t^{K,m}) \right| \le \varepsilon T .$$

*for any $\varepsilon \in (0,1)$ and $h \ge 2 \log T \log \left( \frac{400 \tilde{G} \tilde{\kappa}^{14} \tilde{\kappa}_B^3 \tilde{\kappa}_C^5 \tilde{W}^2 \sqrt{m} d}{\varepsilon \tilde{\gamma}^{9/2}} \log T \log^{1/4} \left( \frac{2}{\tilde{\gamma}} \right) \right)$.*

*Proof.* Since $K \in \tilde{\mathcal{S}}$, there exists a diagonal $L \in \mathbb{R}^{n \times n}$ as in Definition A.8 so that:

$$\mathbf{u}_t^{K,m} = K \mathbf{y}_t^{\mathsf{nat}} + \sum_{i=1}^{m} KC(A + BKC)^{i-1} BK \mathbf{y}_{t-i}^{\mathsf{nat}} = K \mathbf{y}_t^{\mathsf{nat}} + \sum_{i=1}^{m} KCHL^{i-1} H^{-1} BK \mathbf{y}_{t-i}^{\mathsf{nat}} .$$

Then write $L^{i-1} = \sum_{j=1}^{d} \alpha_j^{i-1} e_j e_j^\top$ and obtain

$$\mathbf{u}_t^{K,m} = K \mathbf{y}_t^{\mathsf{nat}} + \sum_{i=1}^{m} KCH \left( \sum_{j=1}^{d} \alpha_j^{i-1} e_j e_j^\top \right) H^{-1} BK \mathbf{y}_{t-i}^{\mathsf{nat}}$$

$$= K \mathbf{y}_t^{\mathsf{nat}} + \sum_{j=1}^{d} KCH e_j e_j^\top H^{-1} BK \sum_{i=1}^{m} \alpha_j^{i-1} \mathbf{y}_{t-i}^{\mathsf{nat}} .$$

Recall $Y_{t-1:t-m}^{\mathsf{nat}} = [\mathbf{y}_{t-1}^{\mathsf{nat}}, \ldots, \mathbf{y}_{t-m}^{\mathsf{nat}}] \in \mathbb{R}^{d \times m}$, define $\boldsymbol{\mu}_\alpha = [1, \alpha, \ldots, \alpha^{m-1}] \in \mathbb{R}^m$ and get

$$
\begin{aligned}
\mathbf{u}_t^{K,m} &= K\mathbf{y}_t^{\mathsf{nat}} + \sum_{j=1}^d KCHe_j e_j^\top H^{-1} BKY_{t-1:t-m}^{\mathsf{nat}} \boldsymbol{\mu}_{\alpha_j} \\
&= K\mathbf{y}_t^{\mathsf{nat}} + \sum_{j=1}^d KCHe_j e_j^\top H^{-1} BKY_{t-1:t-m}^{\mathsf{nat}} \left( \sum_{i=1}^m \boldsymbol{\varphi}_i \boldsymbol{\varphi}_i^\top \right) \boldsymbol{\mu}_{\alpha_j} \qquad \left[ \sum_{i=1}^m \boldsymbol{\varphi}_i \boldsymbol{\varphi}_i^\top = \mathbb{I}_m \right] \\
&= K\mathbf{y}_t^{\mathsf{nat}} + \sum_{i=1}^m \left( \sum_{j=1}^d KCHe_j e_j^\top H^{-1} BK \boldsymbol{\varphi}_i^\top \boldsymbol{\mu}_{\alpha_j} \right) Y_{t-1:t-m}^{\mathsf{nat}} \boldsymbol{\varphi}_i \,.
\end{aligned}
$$

Let $\pi_{h,m,\tilde{\gamma},M^*}^{\mathsf{SC}}$ be the spectral controller with $M_0 = K$ and $M_i^* = \lambda_i^{-1/4} KCH \left( \sum_{j=1}^d \boldsymbol{\varphi}_i^\top \boldsymbol{\mu}_{\alpha_j} e_j e_j^\top \right) H^{-1} BK$ for all $i \in [h]$. Note that we have

$$
\|M_0\| \le \tilde{\kappa}, \quad \|M_i^*\| \le \tilde{\kappa}^4 \tilde{\kappa}_B \tilde{\kappa}_C \cdot \max_{\ell \in [d]} \lambda_j^{-1/4} \langle \boldsymbol{\varphi}_j, \boldsymbol{\mu}_{\alpha_l} \rangle \quad \forall 1 \le j \le m \,,
$$

and from the analysis of Lemma 7.4 of [9], we have that $\lambda_j^{-1/4} \langle \boldsymbol{\varphi}_j, \mu(\alpha_l) \rangle \le \sqrt{\frac{2}{\tilde{\gamma}}}$. Thus, $\|M_i^*\| \le \tilde{\kappa}^4 \tilde{\kappa}_B \tilde{\kappa}_C \sqrt{\frac{2}{\tilde{\gamma}}}$. Then,

$$
\begin{aligned}
\left\| \mathbf{u}_t^{K,m} - \mathbf{u}_t^{M^*} \right\| &= \left\| \sum_{i=h+1}^m KCH \left( \sum_{j=1}^d \boldsymbol{\varphi}_i^\top \boldsymbol{\mu}_{\alpha_j} e_j e_j^\top \right) H^{-1} BKY_{t-1:t-m}^{\mathsf{nat}} \boldsymbol{\varphi}_i \right\| \\
&\le \frac{\tilde{\kappa}^6 \tilde{\kappa}_B \tilde{\kappa}_C^2 \tilde{W}\sqrt{m}}{\tilde{\gamma}} \sum_{i=h+1}^m \sum_{j=1}^d |\boldsymbol{\varphi}_i^\top \boldsymbol{\mu}_{\alpha_j}| \qquad \left[ \|Y_{t-1:t-m}\| \le \frac{\tilde{\kappa}^2 \tilde{\kappa}_C \tilde{W}}{\tilde{\gamma}} \sqrt{m} \right] \\
&\le \frac{30 \tilde{\kappa}^6 \tilde{\kappa}_B \tilde{\kappa}_C^2 \tilde{W}\sqrt{m}}{\tilde{\gamma}^{3/2}} \log^{1/4} \left( \frac{2}{\tilde{\gamma}} \right) \sum_{i=h+1}^m \sum_{j=1}^d \exp \left( -\frac{\pi^2 j}{16 \log T} \right) \qquad \text{[Lemma 7.4 of [9]]} \\
&\le \frac{30 \tilde{\kappa}^6 \tilde{\kappa}_B \tilde{\kappa}_C^2 \tilde{W}\sqrt{m}d}{\tilde{\gamma}^{3/2}} \log^{1/4} \left( \frac{2}{\tilde{\gamma}} \right) \int_h^\infty \exp \left( -\frac{\pi^2 j}{16 \log T} \right) dx \\
&\le \frac{50 \tilde{\kappa}^6 \tilde{\kappa}_B \tilde{\kappa}_C^2 \tilde{W}\sqrt{m}d}{\tilde{\gamma}^{3/2}} \log T \log^{1/4} \left( \frac{2}{\tilde{\gamma}} \right) \exp \left( -\frac{\pi^2 h}{16 \log T} \right) \,,
\end{aligned}
$$

$$
\begin{aligned}
\left\| \mathbf{y}_t^{M^*} - \mathbf{y}_t^{K,m} \right\| &= \left\| \sum_{i=1}^t CA^{i-1} B(\mathbf{u}_{t-i}^{M^*} - \mathbf{u}_{t-i}^{K,m}) \right\| \\
&\le \tilde{\kappa}_B \tilde{\kappa}_C \tilde{\kappa}^2 \sum_{i=1}^t (1-\tilde{\gamma})^{i-1} \left\| \mathbf{u}_{t-i}^M - \mathbf{u}_{t-i}^{K,m} \right\| \qquad \text{[Assumption A.9]} \\
&\le \frac{50 \tilde{\kappa}^8 \tilde{\kappa}_B^2 \tilde{\kappa}_C^3 \tilde{W}\sqrt{m}d}{\tilde{\gamma}^{5/2}} \log T \log^{1/4} \left( \frac{2}{\tilde{\gamma}} \right) \exp \left( -\frac{\pi^2 h}{16 \log T} \right) \,.
\end{aligned}
$$

Using the fact that $\tilde{\kappa}_C \tilde{\kappa}_B \tilde{\kappa}^2 / \tilde{\gamma} > 1$, we get a uniform bound:

$$
\max \left\{ \left\| \mathbf{y}_t^{K,m} - \mathbf{y}_t^{M^*} \right\|, \left\| \mathbf{u}_t^{K,m} - \mathbf{u}_t^{M^*} \right\| \right\} \le \frac{50 \tilde{\kappa}^8 \tilde{\kappa}_B^2 \tilde{\kappa}_C^3 \tilde{W}\sqrt{m}d}{\tilde{\gamma}^{5/2}} \log T \log^{1/4} \left( \frac{2}{\tilde{\gamma}} \right) \exp \left( -\frac{\pi^2 h}{16 \log T} \right) \,.
$$

Whenever $h \ge 2 \log T \log \left( \frac{50 \tilde{\kappa}_B \tilde{\kappa}_C \tilde{\kappa}^2 \sqrt{m}d}{\sqrt{\tilde{\gamma}}} \log T \log \left( \frac{2}{\tilde{\gamma}} \right) \right)$, which is indeed the case for our choice of $\varepsilon$ and $h$, this implies that $\left\| \mathbf{y}_t^{M^*} - \mathbf{y}_t^{K,m} \right\|, \left\| \mathbf{u}_t^{M^*} - \mathbf{u}_t^{K,m} \right\| \le \frac{\tilde{\kappa}^6 \tilde{\kappa}_B \tilde{\kappa}_C^2 \tilde{W}}{\tilde{\gamma}^2}$. Hence, by triangle

inequality $\left\|\mathbf{y}_t^{M^*}\right\|, \left\|\mathbf{u}_t^{M^*}\right\| \leq \frac{4\tilde{\kappa}^6 \tilde{\kappa}_B \tilde{\kappa}_C^2 \tilde{W}}{\tilde{\gamma}^2}$. Thus, the sum of costs is bounded as follows:

$$
\sum_{t=1}^{T} \left| c_t(\mathbf{y}_t^{M^*}, \mathbf{u}_t^{M^*}) - c_t(\mathbf{y}_t^{K,m}, \mathbf{u}_t^{K,m}) \right| \leq \frac{4\tilde{G}\tilde{\kappa}^6 \tilde{\kappa}_B \tilde{\kappa}_C^2 \tilde{W}}{\tilde{\gamma}^2} \sum_{t=1}^{T} \left( \left\|\mathbf{y}_t^{M^*} - \mathbf{y}_t^{K,m}\right\| + \left\|\mathbf{u}_t^{M^*} - \mathbf{u}_t^{K,m}\right\| \right)
$$

$$
\leq \frac{400\tilde{G}\tilde{\kappa}^{14}\tilde{\kappa}_B^3 \tilde{\kappa}_C^5 \tilde{W}^2 \sqrt{m}d}{\tilde{\gamma}^{9/2}} \log T \log^{1/4}\left(\frac{2}{\tilde{\gamma}}\right) \exp\left(-\frac{\pi^2 h}{16 \log T}\right) \cdot T
$$

$$
\leq \varepsilon T. \qquad\qquad [\text{choice of } h]
$$

$\qquad\qquad\qquad\qquad\qquad\qquad\qquad\qquad\qquad\qquad\qquad\qquad\qquad\qquad\qquad\qquad\qquad\qquad\qquad\qquad\qquad\qquad\qquad\qquad\qquad\qquad\qquad\Box$

Using these lemmas, we complete the proof of Lemma A.3 as follows:

*Proof of Lemma A.3.* The proof follows directly from Lemmas A.14 and A.16. $\qquad\Box$

# B  Learning Results

## B.1  Convexity of loss function and feasibility set

To conclude the analysis, we first show that the feasibility set $\mathcal{K}$ is convex and the loss functions are convex with respect to the variables $M$. This follows since the states and the controls are linear transformations of the variables.

**Lemma B.1.** *The set $\mathcal{K}$ from Definition 3.8 is convex.*

*Proof.* Since $\mathbf{x}_t^M, \mathbf{u}_t^M$ are linear in $M$, from the convexity of the norm, the fact that the sublevel sets of a convex function is convex and that the intersection of convex sets is convex, we are done. $\qquad\Box$

**Lemma B.2.** *The loss $\ell_t(M)$ is convex in $M$.*

*Proof.* The loss function $\ell_t$ is given by $\ell_t(M) = c_t(\mathbf{x}_t(M), \mathbf{u}_t(M))$. Since the cost $c_t$ is a convex function with respect to its arguments, we simply need to show that $\mathbf{x}_t^M$ and $\mathbf{u}_t^M$ depend linearly on $M$. The control is given by

$$
\mathbf{u}_t^M = \bar{M}_0 \mathbf{y}_t^{\mathsf{nat}} + \sum_{i=1}^{\tilde{h}} \sum_{j=1}^{\tilde{m}} \lambda_i^{1/4} [\varphi_i]_j \bar{M}_i \mathbf{y}_{t-j}^{\mathsf{nat}} + \sum_{l=0}^{h} \sum_{k=0}^{m} \sigma_l^{1/4} [\phi_l]_k M_{0l} \mathbf{y}_{t-k}^{\mathsf{nat}}
$$

$$
+ \sum_{i=1}^{\tilde{h}} \sum_{j=1}^{\tilde{m}} \sum_{l=0}^{h} \sum_{k=0}^{m} (\sigma_l \lambda_i)^{1/4} [\phi_l]_k [\varphi_i]_j M_{il} \mathbf{y}_{t-j-k}^{\mathsf{nat}}, \qquad\qquad (21)
$$

which is a linear function of the variables. Similarly, the observation $\mathbf{y}_t^M$ is given by

$$
\mathbf{y}_t^M = \mathbf{y}_t^{\mathsf{nat}} + \sum_{q=1}^{t} C A^{q-1} B \mathbf{u}_{t-q}^M.
$$

Thus, we have shown that $\mathbf{y}_t(M)$ and $\mathbf{u}_t(M)$ are linear transformations of $M$. A composition of convex and linear functions is convex, which concludes our Lemma. $\qquad\Box$

## B.2  Lipschitzness of $\ell_t(\cdot)$

The following lemma states and proves the explicit lipschitz constant of $\ell_t(\cdot)$.

**Lemma B.3.** *For any $M, M' \in \mathcal{K}$ it holds that,*

$$
|\ell_t(M) - \ell_t(M')| \leq \frac{32 G \mathcal{R} h \tilde{h} \sqrt{m\tilde{m}} \kappa^4 \kappa_B \kappa_C^2 W}{\gamma^2} \log^{1/2}\left(\frac{2}{\gamma}\right) \|M - M'\|.
$$

*Proof.* Taking the difference in controls defined in eqn. (21):

$$\|\mathbf{u}_t(M) - \mathbf{u}_t(M')\| \leq \frac{16h\tilde{h}\sqrt{m\tilde{m}}\kappa^2\kappa_C W}{\gamma} \log^{1/2}\left(\frac{2}{\gamma}\right)\|M - M'\|$$

Taking the difference in the observations,

$$
\begin{aligned}
\|\mathbf{y}_t(M) - \mathbf{y}_t(M')\| &= \left\|\sum_{q=1}^{t} CA^{q-1}B(\mathbf{u}_{t-q}(M) - \mathbf{u}_{t-q}(M'))\right\| \\
&\leq \frac{\kappa_B\kappa_C\kappa^2}{\gamma} \max_{q\in[t]}\{\|\mathbf{u}_{t-q}(M) - \mathbf{u}_{t-q}(M')\|\} \\
&\leq \frac{16h\tilde{h}\sqrt{m\tilde{m}}\kappa^4\kappa_B\kappa_C^2 W}{\gamma^2} \log^{1/2}\left(\frac{2}{\gamma}\right)\|M - M'\|
\end{aligned}
$$

Using the fact that $\kappa_B\kappa_C\kappa^2/\gamma > 1$, we get a uniform bound:

$$\max\{\|\mathbf{y}_t(M) - \mathbf{y}_t(M')\|, \|\mathbf{u}_t(M) - \mathbf{u}_t(M')\|\} \leq \frac{16h\tilde{h}\sqrt{m\tilde{m}}\kappa^4\kappa_B\kappa_C^2 W}{\gamma^2} \log^{1/2}\left(\frac{2}{\gamma}\right)\|M - M'\|\ .$$

Using the lipschizness of the cost function from Assumption 3.4, the definition of $\mathcal{K}$, we have

$$
\begin{aligned}
|\ell_t(M) - \ell_t(M')| &= |c_t(\mathbf{y}_t(M), \mathbf{u}_t(M)) - c_t(\mathbf{y}_t(M'), \mathbf{u}_t(M'))| \\
&\leq G\mathcal{R}\left(\|\mathbf{x}_t(M) - \mathbf{x}_t(M')\| + \|\mathbf{u}_t(M) - \mathbf{u}_t(M')\|\right) \\
&\leq \frac{32G\mathcal{R}h\tilde{h}\sqrt{m\tilde{m}}\kappa^4\kappa_B\kappa_C^2 W}{\gamma^2} \log^{1/2}\left(\frac{2}{\gamma}\right)\|M - M'\|\ .
\end{aligned}
$$

$\square$

## B.3 Loss functions with memory

The actual loss $c_t$ at time $t$ is not calculated on $\mathbf{x}_t(M^t)$, but rather on the true state $\mathbf{x}_t$, which in turn depends on different parameters $M^i$ for various historical times $i < t$. Nevertheless, $c_t(\mathbf{x}_t, \mathbf{u}_t)$ is well approximated by $\ell_t(M^t)$, as stated in Lemma 4.3 and proven next.

*Proof of Lemma 4.3*: By the choice of step size $\eta$, and by the computation of the lipschitz constant of $\ell_t$ w.r.t $M$ in Lemma B.3, we have:

$$\eta = \frac{\mathcal{R}_M}{L\sqrt{T}},$$

where $L$ is the lipschitz constant of $\ell_t$ w.r.t $M$, computed in Lemma B.3. Thus, for each $j \in [h]$,

$$\|M^t - M^{t-i}\| \leq \sum_{s=t-i+1}^{t} \|M^s - M^{s-1}\| \leq i\eta L = \frac{i\mathcal{R}_M}{\sqrt{T}}\ .$$

Observe that $\mathbf{u}_t = \mathbf{u}_t(M^t)$. Observe that $\mathbf{y}_t$ and $\mathbf{y}_t(M^t)$ can be written as

$$\mathbf{y}_t(M^t) = \mathbf{y}_t^{\text{nat}} + \sum_{q=1}^{t} CA^{q-1}B\mathbf{u}_{t-q}(M^t), \qquad \mathbf{y}_t = \mathbf{y}_t^{\text{nat}} + \sum_{q=1}^{t} CA^{q-1}B\mathbf{u}_{t-q}(M^{t-q})\ .$$

Evaluating the difference,

$$\|\mathbf{y}_t - \mathbf{y}_t(M^t)\| \leq \left\| \sum_{q=1}^{t} CA^{q-1}B(\mathbf{u}_{t-q}(M^t) - \mathbf{u}_{t-q}(M^{t-q})) \right\|$$

$$\leq \kappa_B \kappa_C \kappa^2 \sum_{q=1}^{t} (1-\gamma)^{q-1} \left\| \mathbf{u}_{t-q}(M^t) - \mathbf{u}_{t-q}(M^{t-q}) \right\|$$

$$\leq \frac{16h\tilde{h}\sqrt{m\tilde{m}}\kappa^4 \kappa_B \kappa_C^2 W}{\gamma} \log^{1/2}\left(\frac{2}{\gamma}\right) \sum_{q=1}^{t} (1-\gamma)^{q-1} \left\| M^t - M^{t-q} \right\|$$

$$\leq \frac{16\mathcal{R}_M h\tilde{h}\sqrt{m\tilde{m}}\kappa^4 \kappa_B \kappa_C^2 W}{\gamma\sqrt{T}} \log^{1/2}\left(\frac{2}{\gamma}\right) \sum_{q=1}^{t} q(1-\gamma)^{q-1}$$

$$\leq \frac{16\mathcal{R}_M h\tilde{h}\sqrt{m\tilde{m}}\kappa^4 \kappa_B \kappa_C^2 W}{\gamma^3\sqrt{T}} \log^{1/2}\left(\frac{2}{\gamma}\right).$$

By definition, $\ell_t(M^t) = c_t(\mathbf{y}_t(M^t), \mathbf{u}_t(M^t))$, and by the definition of $\mathcal{K}$ and the projection used in Algorithm 1 we have by Assumption 3.4:

$$\left| \ell_t(M^t) - c_t(\mathbf{y}_t, \mathbf{u}_t) \right| = \left| c_t(\mathbf{y}_t(M^t), \mathbf{u}_t(M^t)) - c_t(\mathbf{y}_t, \mathbf{u}_t) \right|$$

$$\leq GR\|\mathbf{y}_t(M^t) - \mathbf{y}_t\|$$

$$\leq \frac{16GR\mathcal{R}_M h\tilde{h}\sqrt{m\tilde{m}}\kappa^4 \kappa_B \kappa_C^2 W}{\gamma^3\sqrt{T}} \log^{1/2}\left(\frac{2}{\gamma}\right).$$

$\square$

## C   Experiments

We present a series of synthetic experiments designed to evaluate the performance of the DSC, as specified in Algorithm 1. We compare DSC against the gradient response controller (GRC) [30] and the linear quadratic gaussian controller (LQG) [8]. We analyze the performance of these three controllers under the following system settings: a linear signal, where the initial state $\mathbf{x}_0$ is randomly sampled from the Gaussian distribution; and a signal with the ReLU state transition. For each signal type, we evaluate the performance of controllers under two types of perturbations: (i) Gaussian noise, and (ii) sinusoidal disturbances; and provide 95% confidence intervals for each setting.

For each experiment, we consider an LDS with a hidden state dimension of $d = 10$, observation dimension of $p = 3$, and a control dimension of $n = 2$. The system matrix $A \in \mathbb{R}^{d \times d}$ is diagonalizable, with the largest eigenvalue being $0.8$, ensuring marginal stability. The control matrix $B \in \mathbb{R}^{d \times n}$ consists of Normally distributed entries. For the described settings, we use $h = \tilde{h} = 5$ filters and $m = \tilde{m} = 10$ memory for both controllers.

Each figure reports the average quadratic loss computed over a sliding window, with its size set to 10% of the sequence length. Hyperparameters are selected to reflect representative and stable performance, though the experimental setup generalizes to higher-dimensional systems and alternative perturbation models.

The empirical results presented in Figures 3a demonstrate that the DSC controller outperforms the GRC controller, while LQG remains optimal for the setting with Gaussian perturbations. Furthermore, under sinusoidal perturbations, DSC maintains a performance advantage over both LQG and GRC, as illustrated in Figure 3b. Similar conclusions on the performance of the DSC controller hold for the experiments with ReLU state transition as shown in Figures 3c and 3d. The confidence intervals further substantiate that the performance gains of DSC over GRC are consistent and statistically robust across random system initializations.

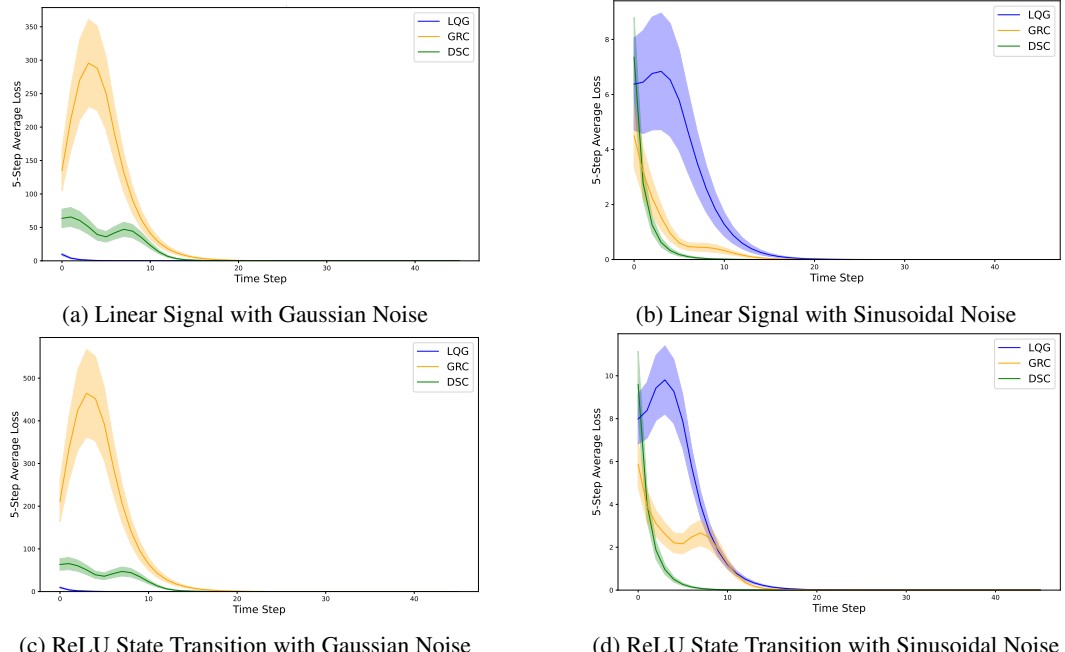

(a) Linear Signal with Gaussian Noise        (b) Linear Signal with Sinusoidal Noise

(c) ReLU State Transition with Gaussian Noise     (d) ReLU State Transition with Sinusoidal Noise

Figure 3: Comparison of Controllers: LQG, GRC, and DSC with 95% Confidence Intervals over 100 trials under Different Input Signal and Perturbation Settings.

## D    Advantage of Smaller $\gamma$

We provide a construction in the fully observed setting, i.e., $C = I$, similar to the one in [9]. Prior work typically assumes a stability margin of $\gamma = \Omega(1/\text{polylog}(T))$ to guarantee an $O(\text{polylog}(T))$ runtime. In contrast, our analysis shows that even when $\gamma = \Omega(1/T^k)$ for any $k \in (0, 1/33)$, one can maintain sublinear regret while still achieving an $O(\text{polylog}(T))$ runtime. This section illustrates that such smaller values of $\gamma$ can lead to a substantially lower aggregate cost compared to the standard choice $\gamma = 1/\text{polylog}(T)$.

Consider a noiseless scalar linear system with parameters $a, b \in \mathbb{R}$ evolving as

$$x_{t+1} = ax_t + bu_t \,.$$

At each time step, the instantaneous loss is defined as

$$\forall t \in [T], \quad c_t(x, u) = \max\{-x, -1\} \,.$$

Since the system is one-dimensional, for sufficiently large $\kappa$, the set of $(\kappa, \gamma)$-diagonalizably stable controllers reduces to

$$S(\gamma) = \{\, k \in \mathbb{R} \mid 0 \le a + bk \le 1 - \gamma \,\} \,.$$

With initial state $x_0 = 1$, the minimum total loss over the horizon $T$ is

$$\min_{k \in S(\gamma)} \sum_{t=1}^{T} c_t(x_t, u_t) = \min_{k \in S(\gamma)} \sum_{t=1}^{T} (-x_t) = -\max_{k \in S(\gamma)} \sum_{t=1}^{T} (a + bk)^{t-1}$$

$$= -\sum_{t=1}^{T} (1 - \gamma)^{t-1} = -\frac{1 - (1 - \gamma)^T}{\gamma} \,.$$

Since $0 \le 1 - \gamma \le e^{-\gamma}$, we obtain the bounds

$$-\frac{1}{\gamma} \le \min_{k \in S(\gamma)} \sum_{t=1}^{T} c_t(x_t, u_t) \le -\frac{1 - e^{-\gamma T}}{\gamma} \,.$$

Applying the lower bound gives

$$\min_{k \in S(1/\mathrm{polylog}(T))} \sum_{t=1}^{T} c_t(x_t, u_t) \geq -\mathrm{polylog}(T) \,,$$

while the upper bound yields

$$\min_{k \in S(1/T^k)} \sum_{t=1}^{T} c_t(x_t, u_t) \leq -T^k(1 - e^{-T^{1-k}}) \leq -\frac{T^k}{2} \,.$$

Therefore, the difference between the two minimum cumulative costs satisfies

$$\min_{k \in S(1/\mathrm{polylog}(T))} \sum_{t=1}^{T} c_t(x_t, u_t) - \min_{k \in S(1/T^k)} \sum_{t=1}^{T} c_t(x_t, u_t) \geq \frac{T^k}{2} - \mathrm{polylog}(T) = \Omega(T^{k/2}) \,.$$

In summary, setting $\gamma = 1/T^k$ results in a markedly smaller cost for the best controller in the class $S(\gamma)$ compared to the $\gamma = 1/\mathrm{polylog}(T)$ case. This demonstrates a polynomial improvement in the achievable aggregate performance.

