# OpenReview forum: "Efficient Spectral Control of Partially Observed Linear Dynamical Systems"
_NeurIPS.cc/2025/Conference — NeurIPS 2025 poster_

### Official Review · Reviewer_tQzd · 2025-06-17

**Clarity:** 3
**Significance:** 2
**Originality:** 2
**Rating:** 4
**Confidence:** 2

**Summary:**

This work studies the problem of online control of partially observed linear dynamical systems, where the cost function is convex and chosen at each time step by an adversary. In this adversarial setting, the authors propose an online optimization algorithm that matches the best known asymptotic regret, while considerably improving the dependency of the computational complexity with respect to the spectral radius. Theorem 4.1 provides the key theoretical results and leads to the claimed complexity bounds. Minimal experiments are included to empirically support the theoretical contributions.

**Questions:**

1. Comparing Equations (1) and (3), shouldn't Line 141 say: “the system would have produced had the learner applied zero disturbances”?

2. What is meant in Line 144 by “natural observations are independent of the learner’s parameters”? How does this enable convex optimization?

3. What does “learning” refer to in this context? Since the system matrices are known and the cost function is defined over known quantities, it is unclear in what sense this is an estimation or learning problem.

4. What is the interpretation of the filter matrix $H$, and how does it help capture long-range dependencies?

5. How could the proposed approach be adapted if the matrices $A$, $B$, and $C$ were unknown?

**Ethical Concerns:**

["NO or VERY MINOR ethics concerns only"]

**Final Justification:**

As someone less familiar with the technical literature in this theoretical subfield of online control, I find the theoretical contributions clear and well executed. I have concerns regarding the applicability and practical motivation of the approach, which I hope the authors might further elaborate on in future revisions or discussions.

I strongly encourage the authors to include a thorough discussion on the modeling choice and the applicability of the studied setup in a revised version of the submission, with simple concrete examples from control.

**Limitations:**

Yes

**Quality:**

3

**Strengths And Weaknesses:**

Overall, the paper is clearly written and presents a significant improvement over prior work in terms of computational efficiency. However, my main concern is the lack of motivation and contextualization of this main contribution, particularly in the introduction and broader discussion.
I would also suggest providing more interpretative insight into the proposed method and to better contextualize the theoretical results by discussing potential practical applications or scenarios where such improvements could be impactful.



### Strengths

- The paper is clearly written, and the mathematical framework is well introduced.
- The related work is appropriately referenced and contextualized.
- The results are clearly presented, with intuitive explanations accompanying the formal statements.
- The theoretical contributions are rigorously derived, with relevant definitions, lemmas, and theorems.
- The limited experiments suggest that for a fixed memory budget, the proposed algorithm outperforms LQG and GRC controllers.
- While I am not a specialist in this subfield, the mathematical arguments appear sound.

### Weaknesses


**Presentation and motivation of the main contribution**

- The main contribution of the paper lies in the improved runtime complexity of the proposed algorithm.
While I recognize that the focus of this work is theoretical, I believe that a brief discussion of the potential implications or contexts where improved complexity would improve the practical relevance of the work. For instance, it remains unclear in which settings compute becomes a primary bottleneck, especially the spectral radius dependence, or how this improvement could concretely benefit applications.
- As noted in the Related Work section, convex parameterizations have been explored in other approaches. However, the paper does not clearly explain how the defined memory-less convex loss compares to others such as GRC. An intuitive explanation of why the proposed relaxation leads to runtime improvements would be valuable.

**Modeling choices**

- The motivation behind the choice of state-space model and convex loss is not sufficiently grounded in real-world examples. Applications are only briefly mentioned in the introduction.
- For example, many classical control problems involve tracking a desired state, where the cost function is a convex function of the state-input pair, $c_t(x, u)$, not the observation-input pair, $c_t(y, u)$. While I understand that the latter allows for theoretical tractability and follows conventions in the literature, it would be helpful to provide some discussion or illustrative examples of how such time-varying convex costs on observations and inputs arise in real-world settings.

**Experimental validation**

- The experimental section is minimal and relegated to the appendix. Key details, such as the explicit form of the time-varying cost function, are missing.
- Importantly, although the computational improvement is the main contribution of the paper, the runtime is not reported in the experiments.
- Including more extensive experiments in the main paper would enhance the impact and help readers assess the practical relevance of the method.



**Technical clarification**

- I believe that there is an issue in the definition of $\gamma$. Indeed, $1-\gamma$ is defined as the spectral radius of the closed loop matrix, which is denoted as $C_\pi$ in the submission, and depends on policy $\pi$. Rather, $1-\gamma$ should be defined as the spectral radius $A$ for the definition to be consistent with the results reported in Table 1.

---

> ### Author Rebuttal · Authors · 2025-07-30
>
> We thank the reviewer for their thoughtful comments and helpful suggestions. We will first address the pointed weaknesses:
>
> **Presentation and motivation of the main contribution.**
>
> Our algorithm improves the runtime complexity of the full control procedure, which we expect to lead to practical runtime gains. Additionally, our method requires fewer parameters, which is often associated with better generalization in statistical learning settings.
>
> We discuss the significance of improved dependence on the stability margin in the introduction (see the paragraph titled *“Marginal Stability and Spectral Filters”*), where we explain that small margins arise in practical systems requiring smooth control inputs such as robotics, thermal regulation systems, and satellite dynamics, where aggressive control is not feasible and long-term memory must be preserved. We will consider expanding this discussion further to strengthen the motivation.
>
> Regarding the choice of loss function: our use of a memoryless convex loss is simply an alternative analysis approach that we believe is clearer than the one in [22], which reduces online control to online convex optimization with memory. Our analysis is more direct and inspired by the approach taken in [15], and we will consider emphasizing this distinction in the camera-ready version. Both works rely on surrogate losses and implicitly manage effective memory.
>
> Our main innovation is not in the loss presentation but in the convex relaxation: by projecting onto the eigenbasis of a particular Hankel matrix, we obtain a controller representation that depends on only $O(\log(1/\gamma))$ terms, as opposed to a polynomial number. This structural insight drives our improved runtime.
>
> **Modeling choices.**
> State-space models with convex loss functions are standard in the control literature and arise in numerous applications (e.g., robotics, economics, mechanical systems). We will consider citing more concrete examples in the introduction.
>
> Regarding the cost structure: since the controller only has access to observations and not the full state, it is not possible to directly minimize a cost function over the full state. While the underlying environment may depend on the full state, regret analysis (in the full-information setting) is meaningful only when the cost functions are defined over the observed quantities. This restriction could potentially be relaxed in a bandit setting, which we leave for future work. We will clarify this point in the paper.
>
> **Experimental validation.**
> We appreciate the suggestion to expand the experiments and agree that it would significantly strengthen the impact and practical relevance of our method. While we agree that demonstrating runtime improvement is important for our work, we chose to include only a preliminary empirical evaluation, intended as a first glimpse into the algorithm’s practical potential.
>
> The cost functions in our experiments are of quadratic form
> $c_t(y_t, u_t) = y_t^\top Q_{\text{obs}} y_t + u_t^\top R u_t$,
> given the observation $y_t$ and the computed control $u_t$ at time step $t$. This choice is made to enable comparison with other methods that only work for quadratic forms.
>
> For the implementing the training update step, we use the memoryless loss
> $\ell_t(M, \bar{M}) = y_t(M)^\top Q_{\text{obs}} y_t(M) + u_t(M)^\top R u_t(M)$
> as in Definition 3.8.
>
> **Technical clarification.**
> The margin $1 - \gamma$ in our work is defined with respect to both the system matrices and the comparator class of LDCs. This is consistent with how the stability margin $\rho$ is defined in [22] (see their Assumption 4).
>
> **Responses to specific questions:**
>
> *Comparing Equations (1) and (3).*
> The learner controls the inputs, while disturbances are adversarially chosen by the environment. Equation (3) is taken directly from Lemma 12.7 in [15], where $y_{\text{nat}}$ represents the output had the learner applied zero control.
>
> *Line 144 (independence of $y_{\text{nat}}$).*
> Because $y_{\text{nat}}$ is generated under zero control, it is independent of the learner’s parameters. This avoids circular dependencies that would otherwise introduce nonconvexities into the problem, e.g., via multiplicative parameter terms.
>
> *Use of “learning.”*
> We agree that this is fundamentally a control problem and will consider emphasizing this framing more clearly. That said, the decision-maker is learning in the sense of approximating the best policy in hindsight.
>
> *Interpretation of the filter matrix.*
> This matrix arises naturally in our analysis and is inspired by recent advances in prediction for linear dynamical systems, where similar Hankel-based constructions are used for convex relaxation. See [16] for context.
>
> *Unknown system matrices.*
> In the unknown dynamics setting, one could first perform system identification and then apply our method. This is similar in spirit to the approach taken in
> *Chen & Hazan (2021), “Black-Box Control for Linear Dynamical Systems,” arXiv:2007.06650*.
> We can mention this extension further in the paper.

---

> ### Comment · Reviewer_tQzd · 2025-08-01
> **Reponse to the authors' rebuttal**
>
> Thank you for your thorough rebuttal.
>
> These clarifications, particularly those concerning the loss minimization formalism, improve understanding the theoretical framing. Including these explanations in the final version would definitely strengthen the paper.
>
> That said, I still find it somewhat difficult to understand the motivation behind focusing specifically on the dependence in the stability margin. While the rebuttal mentions domains such as robotics, it would help to elaborate on when the spectral radius becomes a practical bottleneck in such settings. Are there concrete scenarios where this dependency dominates over other factors, such as model uncertainty or actuation constraints?
>
> Similarly, I appreciate the discussion on the loss structure, but I would be interested in seeing examples, either synthetic or drawn from prior literature, where time-varying convex losses on observations, rather than state, arise in applications. The classical setting that I am aware of in online control is a trajectory optimization cost depending on unobserved states, from which the trajectory and the inputs are optimized for. In this case, the observations are only used to estimate the current state, which is used as an initial conditon for the trajectory optimization problem. How does the framework understudy relate to this setting?
>
> As someone less familiar with the technical literature in this theoretical subfield of online control, I find the theoretical contributions clear and well executed. My questions primarily concern the broader applicability and practical motivation of the approach, which I hope the authors might further elaborate on in future revisions or discussions.

---

> > ### Author Response · Authors · 2025-08-03
> >
> > We thank the reviewer for their thoughtful follow-up.
> > An explicit example illustrating the role of the stability margin appears in Section 9 of [7]. This simple construction demonstrates that systems with long-term memory are best controlled using policies whose closed-loop dynamics have small spectral margin. In such cases, methods with poor runtime dependence on the margin become impractical, highlighting the relevance of our improvement.
> > Regarding the cost’s dependence on the observation: this modeling choice is standard in online control (see Chapter 13 of [15], and also [18, 22]). The key reason is that meaningful regret guarantees are only possible when the comparator policy operates under the same information constraints as the learner. If the comparator had access to the latent state while the learner observes only a partial signal, regret minimization would become vacuous—for instance, when the observation reveals no information (consider C=0, then the observation is identically zero). Thus, the cost must depend on the observation rather than the state to ensure the regret captures performance relative to policies under the same information structure.

---

> > > ### Comment · Reviewer_tQzd · 2025-08-04
> > >
> > > Thank you for your response.
> > >
> > > I understand the necessity for the loss function to depend ont all the available information. However, this modeling choice still seems to blur the line between estimation and control, which are often treated as distinct problems in the literature, such as in [22] where they are shown to be dual. Could the aurhors provide more explicit discussion of this point to avoid potential confusion for readers outside the immediate subfield?
> > >
> > > Additionally, I believe the paper would benefit significantly from a concrete, self-contained example illustrating the practical relevance of the stability margin and why improvements in its associated runtime complexity matter in practice. Since computational efficiency is a central contribution of the work, reporting actual runtimes, at least in a synthetic but illustrative setting, would strengthen the paper’s impact.
> > >
> > > If such clarifying discussions and illustrative examples are provided by the authors, and included in a revised version, I would be inclined to increase my score to 4.

---

> > > > ### Author Response · Authors · 2025-08-04
> > > >
> > > > Thank you for the follow-up and for considering raising your score.
> > > > Regarding the distinction between estimation and control: in [22], estimation of the system matrices is addressed separately, and the resulting method (upon which we build) operates in a control setting with partial observability. Our work assumes known dynamics and focuses entirely on the online control problem, not system identification. We will clarify this distinction and the implications of partial observability in our revision to avoid confusion for readers outside the immediate subfield.
> > > > As for the stability margin, we agree that illustrating its practical relevance is important. Section 9 of [7] provides a concrete analytical example that highlights how runtime complexity depends on the stability margin. We will incorporate a version of this example directly into our paper in a self-contained way.
> > > > Finally, we agree that runtime evaluation would strengthen the empirical component. We will explore the possibility of including additional experiments with runtime comparisons in the camera-ready version.

---

> > > > > ### Comment · Reviewer_tQzd · 2025-08-05
> > > > >
> > > > > My apologies, I meant reference [18], which is a control paper from 1960, not reference [22]. In [18], observations are used to estimate the unknown state.The regulator control problem is adressed while assuming that the state is known.
> > > > >
> > > > > I strongly encourage the authors to include a thorough discussion on the modeling choice and the applicability of the studied setup in a revised version of the submission, with simple concrete examples from control.
> > > > >
> > > > > I increase my score to 4.

---

### Official Review · Reviewer_iEgZ · 2025-06-26

**Clarity:** 3
**Significance:** 2
**Originality:** 3
**Rating:** 4
**Confidence:** 2

**Summary:**

This paper proposes a novel algorithm for controlling linear dynamical systems under partial observation of the latent states and adversarial disturbances. The new algorithm (called DSC) can achieve comparable regret with some prior methods, but has a significant advantage on runtime at eat iteration. The authors show that the per-step runtime of DSC is only $polylog(T/\gamma)$, where $\gamma$  is the stability margin of the system. This improves prior art significantly for small $\gamma$.

**Questions:**

- Can the authors provide examples/situations in practical applications so that $\gamma$ is (very) small?
- What is $poly(\gamma^{-1})$ in each method in Table 1?
- Can we trade off between the regret and per-step runtime?

**Ethical Concerns:**

["NO or VERY MINOR ethics concerns only"]

**Final Justification:**

I have read the authors' response and the other reviews. Most of my concerns have been cleared. I encourage the authors to provide examples/situations in practical applications that may have small stability margin.

**Limitations:**

The authors already discussed some limitations and tried to resolved, e.g. providing some experiments. There seems to be some limitations as pointed out above.

**Quality:**

3

**Strengths And Weaknesses:**

### Strengths:
- The writing of the paper is clear and easy to understand about the problem setting, formulation, and related results.
- The proposed method seems to have a significant advantage on per-step runtime in term of stability margin, but maintain the competitive regret.

### Weaknesses:
- This paper is mostly theoretical with limited validation on practical settings. The authors did a simulation with a simple setup and then compared with two other related methods.
- The main benefit of the proposed method mostly bases on the concept of stability margin. However, the authors have a limited discussion about this concept. This could pose a concern on how practically significant their method is. It could be useful for the readers if the authors can provide examples/situations in practical applications so that $\gamma$ is (very) small. Otherwise, the proposed method may not be practically beneficial.
- Many references lack information.

---

> ### Author Rebuttal · Authors · 2025-07-30
>
> We thank the reviewer for their feedback and for acknowledging the clarity of the paper and the improved runtime dependence on the stability margin. We shall address the weaknesses first, then move on to the questions raised by the reviewer.
>
> **Weaknesses**
> **Experiments:** We appreciate the reviewer’s observation regarding the scope of the experimental evaluation. Our primary aim in this work was to highlight and validate the theoretical contributions under well-controlled and interpretable conditions. That said, we fully acknowledge the importance of broader empirical validation, especially in settings that reflect more diverse and practical scenarios. While the current experiments are limited in scope, we view this as a foundation for future work that will explore a wider range of system dynamics and environments to further demonstrate the robustness and applicability of our approach.
> **Stability margin:** We kindly refer the reviewer to Section 9 of [7], which discusses settings where the stability margin is small. We also mention the benefits of small $\gamma$ in the introduction (see the paragraph titled *“Marginal Stability and Spectral Filters”*), and we will consider expanding this discussion. In general, our method is beneficial in situations where the system exhibits long memory and must be controlled smoothly. Examples include structural damping in buildings, thermal regulation systems, and high-precision automated surgery, among others.
> **References:** We will expand and improve the descriptions of referenced works throughout the paper to ensure clarity and completeness.
>
> **Questions**
> **Examples of small $\gamma$:** Please see our response above and Section 9 in [7].
> **$\gamma$ dependence in Table 1:** In our work, the runtime exponent in $\gamma$ is 11. In [22], the dependence is noted as polynomial but the exact power is not specified and is nontrivial to extract from their analysis. That said, our focus is not on the specific power but on the qualitative improvement: we achieve standard $\widetilde{O}(\sqrt{T})$ regret while reducing the runtime dependence on $\gamma$ from exponential to polynomial.
> **Tradeoff between regret and runtime:** We are not aware of a way to achieve such a tradeoff within our current framework. Our regret bound is already satisfactory given the complexity constraints, but exploring this tradeoff is an interesting direction for future work.

---

> > ### Comment · Reviewer_iEgZ · 2025-08-06
> >
> > I thank the authors for their responses which mostly clear out my concerns.

---

### Official Review · Reviewer_DfSy · 2025-06-26

**Clarity:** 3
**Significance:** 2
**Originality:** 3
**Rating:** 4
**Confidence:** 5

**Summary:**

The paper proposes a new method for online control of partially observed linear systems that is amenable to efficient online updates. The setting of the paper is that of online non-stochastic control where the costs and (potentially adversarial) disturbances are revealed sequentially.
The main challenge with previous approaches ([22]) is that the run-time per step (one online update) scales inversely proportially ($O(1/\gamma)$) with the stability margin $\gamma$ of the system, as the margin approaches zero. In particular, the number of tunable parameters scales polynomially with $1/\gamma$.  By following the proposed approach the run-time scales with the improved rate of $O(\log(1/\gamma))$, which is an exponentially better for very small $\gamma$. In the meantime, we preserve the standard $O(\sqrt{T})$ regret guarantees of the convex cost setting (where $T$ is the total number of steps taken).

The main idea is to pre-filter (convolution operator) the observations twice with appropriate filters that are selected as the top eigenvectors of specific Hankel matrices. Then, the controller is obtained as a (convex) linear combination of the pre-filtered observations,where the linear coefficients are tuned online. Under certain conditions on the system and controller class, these specialized Hankel eigenvectors compress the information of the observations. As a result, it is sufficient that the number of tunable parameters scale only logarithmically with $1/\gamma$.

These conditions require the system to be stable, diagonalizable, with positive real eigenvalues in both open-loop and closed-loop.

**Questions:**

**Major comments**

I am willing to improve my evaluation based on the authors response. To summarize, the limitations should be discussed more explicitly, comparison with prior work should be expanded, the literature review should be enhanced and some aspects of the analysis/contribution should be clarified.

**1.** Dealing with weakness 1. In the limitations, it should be acknowledged that this assumption is unrealistic for actual physical systems. If the authors provide an example of a meaningful real-world physical system that satisfies the conditions I will improve my rating even further. The simulation should also include examples of systems that violate the conditions (in both open and closed loop) to support the claim of the footnote. In any case, there should be more discussion. Why are these assumptions needed? Is it an artifact of the proof? A fundamental limitation? What breaks in the case of complex or negative eigenvalues or in the case of Jordan blocks? Would we need a different Hankel matrix?

**2.** Dealing with weakness 2. It should be clarified if the order of $1/\gamma^{11}$ in Theorem 4.1 is worse/better/same compared to [22].  From this perspective, the claim "optimal regret guarantees" in the caption of Table 1 is currently unclear.

**3.** Dealing with weakness 3. The paper would benefit from the addition of more papers from the control community. For example, the setting of Hinfinity control where the controller acts against adversarial disturbances seems quite relevant. Variations of this idea include mixed H2/Hinfty control, risk sensitive control, regret optimal control, minimax adaptive control. The following papers might be relevant and contain useful references

T. Basar and P. Bernhard, $\mathcal{H}_{\infty}$, "Optimal Control and Related Minimax
Design Problems — A dynamic Game Approach", 2nd ed. Boston, MA,
USA: Birkhauser, 2008

Kaiqing Zhang, Bin Hu, Tamer Basar, "Policy Optimization for $\mathcal{H}_2$ Linear Control with $\mathcal{H}\_{\infty}$ Robustness Guarantee: Implicit Regularization and Global Convergence", SIAM, 2021

G. Goel and B. Hassibi, "Regret-Optimal Estimation and Control," in IEEE Transactions on Automatic Control, vol. 68, no. 5, pp. 3041-3053, May 2023

O. Kjellqvist and A. Rantzer, "Output Feedback Minimax Adaptive Control", arxiv, 2024

**4.** Dealing with weakness 4. The paper could provide more intuition on how spectral filters work. I had to go to [15] to understand that the Hankel eigenvectors compress information. I recommend to include a brief discussion here as well.

It could also help to provide more intuition on what the main challenge in [22] is. Why does the run-time scale with $1/\gamma$? It seems that a crucial component is studying the inequality $(1-\gamma)^m\le c$, where $m$ is the number of parameters and $c$ can be arbitrary. This inequality is equivalent to $m\ge -\log c/\log(1-\gamma)$. Although there is a logarithm, as the margin goes to $0$, we have $-1/\log(1-\gamma)\sim 1/\gamma$. I think the paper would benefit from such a discussion.

**5.** Dealing with weakness 5. Please explain why we need double filtering in more detail.

**6.** Convolution implementation. A naive implementation of the prefiltering steps would lead to a run-time that is polynomial with $1/\gamma$. The reason is that naive convolutions involve $m,\tilde{m}$ mutliplications. In turn, $m,\tilde{m}$ scale polynomially with the inverse stability margin. The paper claims that this limitation can be lifted by implementing the convolutions efficiently.
However, the paper does not provide enough details to understand how this is done. Unfortunately, the proof of Corollary 4.4 is informal and handwavy. Could you please provide more details? I suggest to make the proof of Corollary more precise by referring to specific results in [2] in a more principled way.

**7.** Please provide more details about the simulations. Does matrix A satisfy all the restrictive stability assumptions including positive real eignevalues? Have you tried a matrix A that does not? What is the frequency of the sinusoid. Have you tried several frequencies? (Otherwise, it might be a matter of luck that the controller works well with a specific frequency.)

**8.** In line 129 the following is stated "– Appendix C reports preliminary empirical evaluations supporting our theoretical findings". The theoretical results only claim that the proposed method has superior runtime. They do not claim anything regarding the performance or the regret order. The simulation results show that the proposed method is superior compared to [22] but they do not compare runtimes. Hence, I am quite confused since the theoretical claims and the simulations findings are about different things.

**Other comments**
**i)** Can we have measurement noise?

**ii)** I couldn't find the definition of $\mathcal{A}_{CL}$ (mentioned in line 194) anywhere. Could you clarify the definition?

**iii)** What is the difference between frequency domain transformations and spectral filtering? Maybe the paper could include some discussion in the related work.

**iv)**. I think $\tau$ and $t$ mixed up in several steps in the proof of lemma A.2, e.g., between 746-747.

**v)**. Please define $y_t(M)$ (line 216) formally. The current phrasing is ambiguous.

**vi)**. Is $y^{nat}$ missing from $y_t(M^t)$, $y_t$ below line 882? In general the notation for the several versions of $y$ is quite confusing.

**Ethical Concerns:**

["NO or VERY MINOR ethics concerns only"]

**Final Justification:**

Although the weaknesses remain (restrictive assumptions, limited simulations), the authors have committed to clarifying their limitations in more detail. From this perspective, I am increasing my score. To summarize, the theoretical result is quite interesting and could inspire interesting future work directions where these limitations could be addressed. Moreover, I feel that overall the paper is well-written. Hence, the benefits outweigh the disadvantages.

**Limitations:**

As I describe in the weaknesses and questions the limitations should be further clarified as they are only partially addressed. (There are only technical limitations no societal issues.)

**Paper Formatting Concerns:**

no concerns

**Quality:**

3

**Strengths And Weaknesses:**

**Strengths**

The idea of pre-filtering observations to improve dependencies on the stability margin is reasonable and interesting.  While I wouldn't rank it as the most important practical aspect it has theoretical interest. It can be impactful when the system exhibits long memory and slow attenuation. The analysis seems sound and correct. The strongest point of the paper is that the number of tunable controller parameters scales logarithmically with the inverse stability margin. The idea of pre-filtering has been studied before in prior work, e.g. in [7] for fully observed systems. Interestingly, here double filtering is used here for partially observed systems which is a non-trivial change.

**Weaknesses**
The results are quite preliminary and lack generality. Moreover, little intuition is provided about why the method works.
Based on the responses I am willing to improve my evaluation. To simplify the authors' response I included the actionable items in the questions section.

**1.** Assumption 3.5 and Definition 3.4 seem extremely restrictive. They require the following: First, the controller itself is required to be implemented as a stable diagonalizable system with positive real eigenvalues. Second, the overall closed-loop dynamics should be stable diagonalizable with positive real eigenvalues. Third and perhaps more importantly, Assumption 3.5 requires the open-loop dynamical system itself to be stable, diagonalizable and to have positive real eigenvalues.
The limitations section should be expanded to discuss this in more detail. Currently, only issue 1 is discussed, while issue 3 is hidden; I only understood it while going through the proofs.

Assuming stable and diagonalizable is ok. However, assuming systems with positive and real eigenvalues (both in open-loop and in closed-loop) is very peculiar. I have doubts that this can include meaningful systems. Almost all physical systems of interest exhibit oscillations, meaning they have complex eigenvalues. Even physical systems with positive real eigenvalues, like higher order integrators (which are excluded since they are not diagonalizable), will have complex eigenvalues in closed-loop. I couldn't find any evidence supporting footnote  1 "This requirement is for tractability of analysis alone, and we show empricially that our method works regardless of it".

**2.** The paper does not compare the regret order (with respect to $\gamma$) with that of prior work. It only compares run-times.

**3.** The literature review is very biased with 16/24 references containing the same author. The paper lacks references to works from the control community.

**4.** There is little intuition about how spectral filters work.

**5.** The paper does not explain what the difference with [7] is and why we need double filtering instead of single.

---

> ### Author Rebuttal · Authors · 2025-07-30
>
> We thank the reviewer for his thorough reading of our paper and proofs and giving educated comments about our work.
>
> **Response to Comment on Definition 3.4 and Assumption 3.5**
> We thank the reviewer for raising this important point.
> Definition 3.4 defines the comparator class, and we agree that the assumption that it must include the zero policy is a restrictive modeling choice. We will add this to the limitations section.
> Indeed, Assumption 3.5 applies only to the comparator class, not to the learned policy itself. This assumption is an artifact of the analysis. Our theoretical regret bounds rely on this structure, but our experiments optimize loss rather than regret, and thus are independent of the comparator class. The fact that our algorithm performs well in practice even without satisfying Assumption 3.5 supports the view that this assumption is not fundamental to performance.
> That said, the requirement for real positive eigenvalues stems from the spectral filtering technique itself. The filters we use are based on Hankel matrices, whose structure relies on real, monotonic dynamics; complex eigenvalues would yield oscillatory behavior that breaks this structure. Therefore, this assumption reflects a fundamental limitation of our current approach. However, it may be possible to overcome it using recent advances such as those in:
> Marsden & Hazan (2025), *Universal Sequence Preconditioning*, arXiv:2502.06545.
> We will consider discussing this direction in the camera-ready version.
> Finally, we note that it is possible to relax the non-negativity requirement on eigenvalues and extend our analysis to systems with marginally stable real eigenvalues. In this case, the integration would start from $-1 + \gamma$ rather than 0. We will clarify this technical point in a footnote.
>
> **Response to Comment on Regret Bound and Comparison with [22]**
> We appreciate the reviewer’s request for clarification on the regret dependence in our work versus [22].
> In our setting, the regret scales as $\widetilde{O}(\sqrt{T})$, and the number of tunable parameters scales as $O(\log(1/\gamma))$, where $\gamma$ is the stability margin. In [22], while the regret is also $\widetilde{O}(\sqrt{T})$, the number of parameters scales polynomially with $1/\gamma$. Unfortunately, [22] does not specify this exponent precisely, and it is nontrivial to extract from their analysis.
> Our key contribution is that we achieve the same optimal regret dependence on $T$ while significantly improving the complexity dependence on the stability margin $\gamma$ — from polynomial to logarithmic in the number of parameters and runtime. This is the basis for our use of the term “optimal regret” in Table 1, though we very much agree that the wording should be improved for clarity and will revise the paper accordingly.
>
> **Response to Comment on Related Work**
> We agree that the related work section is currently too narrow and appreciate the reviewer’s suggestions. We'll revise the section to better reflect contributions from the control community and include a more diverse set of references in the camera-ready version, including the ones mentioned by the reviewer.
>
> **Response to Comment on Spectral Filtering Intuition**
> We leverage the fact that the controllers in a LDC have the structure of convolution of the natural observations with vectors of the form $[1, \alpha, \alpha^2, \ldots]$, and spectral filters provide a compact, universal representation for such sequences. This structure allows for an efficient low-dimensional approximation using the top eigenvectors of Hankel matrices. We will add a brief explanation of this intuition in the paper.
> Regarding the comparison to [22], the key inequality used there is $(1 - \gamma)^m \leq e^{-\gamma m}$, which implies $m \geq \log(c)/\gamma$. This results in polynomial dependence on $1/\gamma$. Our approach bypasses this by using a more refined analysis based on Hankel matrices, which leads to logarithmic dependence in the number of parameters. We will make this contrast clearer in the final version.
>
> **Response to Comment on Double Filtering**
> Our method consists of two stages: lifting and learning. Spectral filtering is applied in both. The lifting stage enables us to define a comparator class with desirable structure; the learning stage operates in the lifted space. If we only applied filtering at the learning stage, following a standard lifting approach as in [15], the ambient dimension would still be $O(1/\gamma)$, so spectral filtering would not reduce the number of parameters. Therefore, double filtering is necessary to obtain the improved dependence on $\gamma$.
> It may be possible to avoid explicit lifting with an alternative analysis and assumptions by analyzing a complicated multiplication of matrices, but this would diverge significantly from the structure in prior work. We found our approach to be more direct and conceptually aligned with the literature. We will clarify this point in the paper.
>
> **Response to Comment on Corollary 4.4**
> We agree that Corollary 4.4 can be clarified. The key observation is that we compose two convolutions. Theorem 3 in [2] guarantees that each such convolution can be computed efficiently. We will revise Corollary 4.4 to explicitly cite this result and make the argument more rigorous.
>
> **Response to Comment on Experiments**
> We agree with the reviewer that including a broader set of experiments, both with a wider range of noise frequencies and with system matrices that do not satisfy the current stability assumptions, could provide additional insight and further test the robustness of our approach.
> For all conducted experiments, we ensured that the system matrix $A$ is symmetric with real eigenvalues between 0 and 0.8.
> We experimented with random frequencies of the sinusoidal noise and observed generally consistent controller behavior. For the results shown in Figure 3, a frequency of 0.1 was selected as it most clearly illustrated the observed patterns.
>
> **Response to Comment on Theory vs. Experiments**
> We agree that the theory focuses on runtime improvements, while the experiments evaluate loss. We will make this distinction clearer. Although we do not claim theoretical improvements in regret, the improved empirical performance may be explained by the fact that our method captures a longer effective history for the same number of parameters. We will add this discussion to the experimental section.
>
> **Other Comments**
> i) Our analysis holds for bounded measurement noise as well. Adding noise to the measurements is equivalent to adding additional noise to the already existing disturbance. Since we use the disturbance in the definition of $y_{\text{nat}}$, and as long as those are bounded, the analysis remains valid. We will add this clarification to the paper.
> ii) We chose to define $A_{\text{CL}}$ implicitly, as its matrix form is very complex. It represents the closed-loop matrix of the policy defined in Lemma 2, and is effectively the induced matrix appearing in Definition A.8, corresponding to the spectral-lifted policy. We will consider whether it is feasible to write the definition explicitly.
> iii) The Fourier basis corresponds to a different set of eigenfunctions, and we are not aware of any logarithmic approximation guarantees using them. The guarantees we rely on are specific to the eigenvectors of the Hankel matrix. The reason is that the Fourier basis is orthonormal, i.e., all eigenvalues are one, whereas the SF basis has an exponential eigenvalue decay. This is a crucial point, and we will try to emphasize it in the camera ready version as well.
> iv) We thank the reviewer for pointing out the typo in Lemma A.2. The subscript in line 746 should be $\tau - 1$, and the superscript should be $\pi_\tau$. The summation should also go up to $h$. This follows from the definition of the policies $\pi_\tau$, and line 747 then follows correctly.
> v) $y_t(M)$ is defined by plugging $u_t(M)$ into the LDS equation (1), where $u_t(M)$ is defined in Definition 3.6. We will formally define $y_t(M)$ in the paper as well.
> vi) The reviewer is also correct about the final typo. We will fix this in the camera-ready version; those terms cancel out.

---

> > ### Comment · Reviewer_DfSy · 2025-08-03
> > **Response to rebuttal by authors**
> >
> > Thank you for the response. The authors seem committed to explaining several limitations of the paper more clearly. If this is the case I will increase my score to 4. Some concerns remain.
> >
> > "Assumption 3.5 applies only to the comparator class, not to the learned policy itself". Sure but then the main regret guarantee relies on this comparator class. So it is possible that the regret guarantees are not very meaningful if the comparator class is too restrictive.
> >
> > "This assumption is an artifact of the analysis." I am not convinced that this is the case since as the rebuttal explains we would need a different Hankel structure to analyze the complex case.
> >
> > "The fact that our algorithm performs well in practice even without satisfying Assumption 3.5": I am confused about this since in the response this is mentioned "For all conducted experiments, we ensured that the system matrix $A$ is symmetric with real eigenvalues between 0 and 0.8." It seems that the experiments are for matrices that satisfy the assumptions. Have you tried systems that are more general? Otherwise how can this claim be supported?

---

> > > ### Author Response · Authors · 2025-08-03
> > >
> > > We thank the reviewer for the follow-up and for considering raising their score. We are indeed committed to explaining the paper’s limitations more clearly in the camera-ready version.
> > > We agree that while Assumption 3.5 applies only to the comparator class, it directly impacts the regret bound, and extending to a more expressive class would likely require a different analysis. This assumption is fundamental to our current proof technique, rather than a superficial artifact.
> > > Regarding experiments: our point was that they measure cumulative loss, not regret, so Assumption 3.5 does not affect them. That said, all experiments used systems satisfying the assumption, and we agree this does not justify it. We will clarify this and avoid relying on the experiments to support the assumption.

---

> > > > ### Comment · Reviewer_DfSy · 2025-08-03
> > > > **Acknowledgment**
> > > >
> > > > Thanks a lot for the clarifications

---

### Decision · Program_Chairs · 2025-09-17

**Decision:**

Accept (poster)

**Comment:**

This paper proposes a new method for controlling partially observed linear dynamical systems with known dynamics. In the considered setting, costs and potentially adversarial disturbances are revealed sequentially, and the goal is to design an efficient algorithm that also achieves desirable regret guarantees. The authors’ proposed method successfully accomplishes both.

Three reviews were received. All reviewers agreed that the paper provides interesting results and introduces a novel methodology. One reviewer noted that the idea of pre-filtering observations to improve dependencies on the stability margin is particularly interesting. The reviewers also confirmed that the paper is well written. On the critical side, however, some highlighted potentially restrictive assumptions (Assumption 3.5 and Definition 3.4). In addition, they noted that more references to relevant work in the control community would strengthen the paper, and they questioned the practical relevance of the proposed method given the very preliminary nature of the experiments.

Overall, the AC concurs with the reviewers that the strengths of the paper outweigh its weaknesses, but strongly encourages the authors to address the reviewers’ comments in their revision, as promised.